# Comparative characterization of Cas12f orthologs reveals mechanistic features underlying enhanced genome editing efficiency

Miniature CRISPR–Cas12f nucleases are attractive candidates for therapeutic genome editing because of their compact size and compatibility with adeno-associated virus (AAV) delivery. However, editing efficiencies in mammalian cells are lower than those of larger systems. The extensive phylogenetic diversity of Cas12f suggests unexplored mechanistic variation with the potential for optimization. Here we identify and characterize a naturally occurring Cas12f ortholog discovered through metagenomics, *Alistipes* sp. Cas12f (Al3Cas12f), which supports robust genome editing in human cells. Through structural, biochemical and kinetic analyses, we compare Al3Cas12f to two recently described orthologs, *Oscillibacter* sp. Cas12f and *Ruminiclostridium herbifermentans* Cas12f. These orthologs present divergent architectures and regulatory features governing protospacer-adjacent motif recognition, guide RNA (gRNA) binding, dimerization and DNA cleavage. Notably, Al3Cas12f achieves efficient R-loop formation through a stable dimer interface and a naturally optimized gRNA. Leveraging these structural insights, we generate an engineered Al3Cas12f variant (RKK) that increases editing and improves activity across several tested genomic loci. By overcoming locus-dependent variability and an apparent potency threshold, this engineered compact editor seems to expand the feasibility of low-dose, AAV-compatible therapeutic genome editing. Our results elucidate mechanistic determinants of Cas12f activity and offer a framework for engineering compact genome editors that may bear therapeutic potential.

CRISPR–Cas systems that provide adaptive immunity against mobile genetic elements in prokaryotes have been repurposed as genome-editing tools in various organisms[1–5]. These highly diverse CRISPR–Cas systems are divided into two main classes (class 1 and class 2), each containing several distinct types[6]. Class 2 effectors, such as Cas9 (type II), Cas12 (type V) and Cas13 (type VI), use a single effector protein as a nuclease. Among these, *Streptococcus pyogenes* Cas9 (SpCas9) and *Acidaminococcus* sp. Cas12a (AsCas12a) have robust nuclease activity and are widely used for genome engineering[1,7–10]. However, both proteins are relatively large (exceeding 1,300 aa), which presents challenges for packaging into adeno-associated virus (AAV) vectors and mRNA manufacturing, resulting in inefficient delivery and limiting their therapeutic applications[11].

To overcome limitations in AAV packaging and delivery, smaller CRISPR–Cas subtypes have emerged as candidates for genome editing[12–14]. One such subtype, type V-F (Cas12f) nucleases (ranging

✉e-mail: dtaylor@utexas.edu

from 400 to 700 aa)[15], target double-stranded DNA (dsDNA) with 5′ T-rich or C-rich protospacer-adjacent motif (PAM) preferences[13,16,17]. Cryo-electron microscopy (cryo-EM) studies revealed that Cas12f functions as an asymmetric homodimer, where one monomer cleaves both the target DNA strand (TS) and the nontarget DNA strand (NTS), while the other remains inactive[18–23]. Although these proteins have yet to reach saturating levels of editing in mammalian cells, rational engineering of Cas12f proteins, as well as their gRNA scaffolds, have greatly enhanced their gene-editing activity[21,23–26].

Continuous discovery of new Cas12f variants underscores the diversity of type V-F nucleases and the potential for uncovering additional naturally optimized orthologs with high intrinsic activity. In this study, we present a newly identified Cas12f ortholog from *Alistipes* sp., Al3Cas12f, which exhibits high editing efficiency in human cells. To contextualize the properties of Al3Cas12f, we also studied two newly identified Cas12f orthologs that originate from *Oscillibacter* sp. (OsCas12f) and *Ruminiclostridium herbifermentans* (RhCas12f). Both orthologs exhibit high editing efficiency in human cells[17]. We conducted kinetic analyses of all three orthologs and performed direct comparisons of structural and biochemical features. Distinct mechanisms for gRNA binding, PAM recognition and DNA targeting were elucidated. Comparison across Cas12f orthologs expanded the structural and mechanistic framework for type V-F enzymes and, in turn, enabled structure-guided engineering of Al3Cas12f to enhance activity across tested targets, supporting therapeutic development of compact genome editors.

## Results

### Al3Cas12f is a highly active CRISPR–Cas nuclease

To explore the diversity of type V nucleases, we mined a large database of high-quality assembled microbial metagenomes from diverse environments, uncovering thousands of genes encoding small nucleases in the vicinity of CRISPR arrays. Although divergent, some of these nucleases are related to other Cas12f nucleases when placed in a phylogenetic tree of archetypal type V nucleases (Fig. 1a). Newly identified representative sequences Cas12f-MG119-1, Cas12f-MG119-2, Cas12f-MG119-3 and Al3Cas12f share 28–52% average amino acid sequence identity with reference sequences and range between 433 and 488 aa in length (Fig. 1a and Supplementary Data 1). Upon inspection of the genomic regions encoding the nuclease genes, we identified the corresponding CRISPR RNAs (crRNAs) and *trans*-activating crRNAs (tracrRNAs). Cas12f-MG119 representatives require large gRNAs (129–156 nt), as empirically confirmed (Supplementary Fig. 1 and Supplementary Data 1). We determined that these systems recognize T-rich PAMs using an in vitro cleavage assay of an 8N PAM library and cut the TS at 20, 22 and 23 nt from the 5′ PAM sequence and the NTS at 11 and 13 nt (Fig. 1b, Supplementary Fig. 2 and Extended Data Fig. 1). Size-exclusion chromatography (SEC) of the purified protein revealed that Al3Cas12f exists as an obligate dimer even in the absence of any gRNA (Extended Data Fig. 2).

We next studied the activity of Al3Cas12f in human cells. Results from a gRNA screen targeting intron 1 of the *ALB* gene, exon 3 of the *APOA1* gene and the AAVS1 site within *PPP1R12C* intron 1 showed that 27 target sites displayed >10% editing, 19 sites displayed >50% editing and 10 sites displayed >90% editing across AAVS1 and *APOA1* (assessed as the percentage of reads from next-generation amplicon sequencing that contained insertions or deletions (indels); Fig. 1c). In addition to screening Al3Cas12f across multiple loci we also tested different spacer lengths to optimize Al3Cas12f cleavage activity. Results revealed that a 19-nt spacer length seemed optimal across multiple sites, with the AAVS1-F1 gRNA showing similar editing efficiencies across 19–21-nt spacers (Extended Data Fig. 3). Together, these results suggest that Al3Cas12f is highly active in its native form. To further validate the high levels of activity observed in mammalian cells, we compared the activity of Al3Cas12f relative to other CRISPR–Cas orthologs. Al3Cas12f

outperformed OsCas12f across six of eight loci and exhibited significantly higher activity than LbCas12a at two sites (*NLRC4* and *NUDT16*) (Fig. 1d,e). These results indicate that this nuclease is a valuable candidate for structural and biochemical characterization relative to other Cas12f systems.

### Cryo-EM structures of Al3Cas12f, OsCas12f and RhCas12f ternary complexes reveal differences in dimer interface

To understand the structural basis for enhanced activity of Al3Cas12f relative to other orthologs, we first solved a cryo-EM structure of the enzyme bound to its gRNA targeting a 55-bp target dsDNA. The Al3Cas12f structures revealed an asymmetric homodimer in a post-cleavage state. The dimer is anchored by multiple interactions within its recognition (REC) domains, as well as direct base-stacking interactions with the gRNA scaffold and the R-loop. Consistent with previously solved Cas12f structures, the model revealed a bilobed ternary ribonucleoprotein (RNP) complex with the recognition lobe comprising the REC and wedge (WED) domains, while the nuclease lobe contains both RuvC and ZF domains (Fig. 2a, Extended Data Figs. 4 and 5 and Table 1). This enzyme most closely resembles the structure of *Clostridium novyi* (CnCas12f), with REC domains comprising large extended α-helices[22]. However, there are notable differences between the structures of these enzymes. Al3Cas12f REC domains directly contact stem 2 of its gRNA, as opposed to CnCas12f, which has a shorter stem 2. Our structure contains a fully formed R-loop with both RuvC domains docked, as opposed to the precleavage complex observed in CnCas12f, which forms a partial R-loop (Supplementary Fig. 3). Because of these differences, we sought to obtain a better mechanistic understanding of Cas12f rearrangements as a function of R-loop formation.

As there are limited Cas12f structures available, we aimed to perform a more thorough comparison of several orthologs known to have high levels of genome-editing activity. A recent publication highlighted the use of OsCas12f and RhCas12f as potential genome-editing tools[17]. Therefore, we reasoned that obtaining structures of OsCas12f and RhCas12f bound to their respective gRNAs and target DNA might provide more insights into the mechanism of Cas12f nucleases (Fig. 2b,c, Extended Data Figs. 4 and 5 and Table 1). While all structures showed bilobed asymmetric homodimers with a fully formed 20-bp R-loop, RhCas12f and OsCas12f contained notably smaller REC domains relative to Al3Cas12f (Fig. 2b,c). In addition, only Al3Cas12f and OsCas12f had resolvable RuvC domains tethered to the R-loop, while neither RuvC domains were resolved for RhCas12f (Fig. 2c). These observations suggest that dimerization interfaces vary widely among Cas12f nucleases and that the RuvC domains can sample multiple conformational states and exhibit different levels of structural plasticity across Cas12f orthologs.

### Cas12f gRNAs exhibit structural and functional diversity

The Cas12f ternary complexes demonstrate notable structural diversity among the cognate gRNAs of these nucleases. The gRNA scaffolds of OsCas12f and RhCas12f share a conserved L-shaped architecture (Fig. 3a,b), like AsCas12f. However, their gRNAs are much shorter than that of their counterpart, whereby they contain only four stem loops, as opposed to the five previously described in AsCas12f (Extended Data Fig. 6a,b).

The long stem 2 of the gRNA of RhCas12f and OsCas12f is nearly perpendicular relative to stems 3 and 4. While this region is not well resolved for OsCas2f, stem 2 in the RhCas12f gRNA exhibits a 90° kink that creates additional stabilizing contacts with the Cas12f monomer 2, making it highly resolvable in our structure (Fig. 3b). This region docks into the groove formed by the REC and WED domains and contacts the PAM-interacting residues of monomer 2. Notably this region of the gRNA shares the same sequence with its PAM and is recognized by the same residues involved in PAM binding by monomer 1. Specifically,

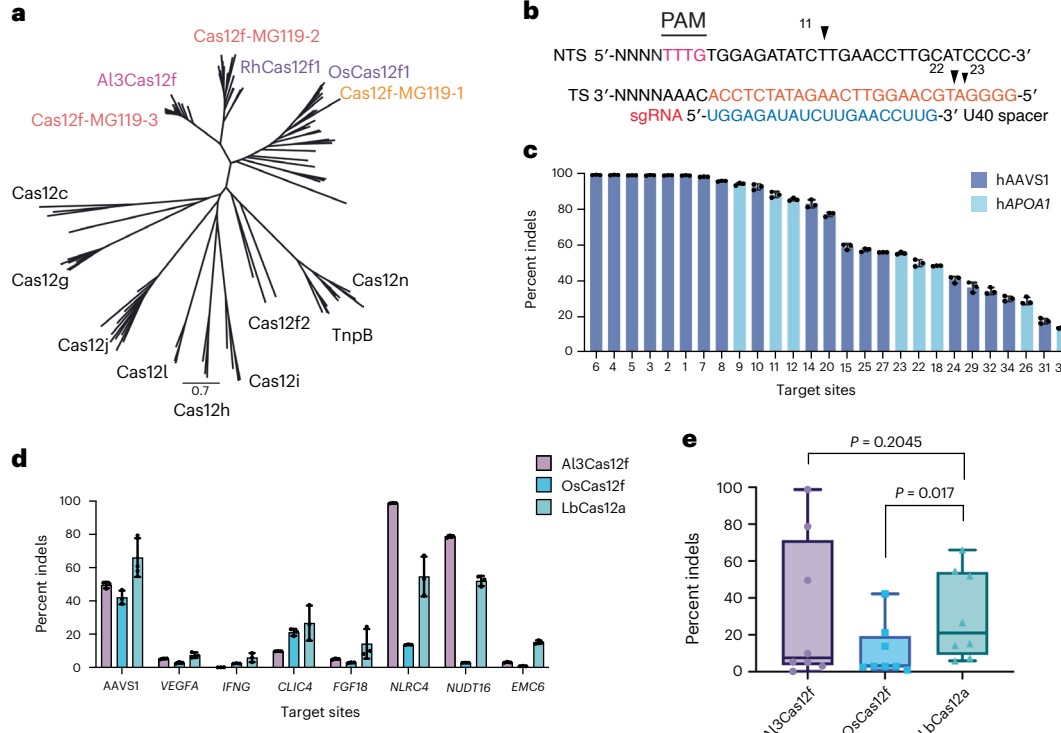

**Fig. 1 | Initial discovery and characterization of the compact type V Al3Cas12f nuclease. a**, Phylogenetic tree showing representative members of the Cas12f-MG119 family of compact type V systems (including the Al3Cas12f clade) in relation to other Cas12 and TnpB nucleases in the literature. **b**, Schematic representing the cut sites for Al3Cas12f on the TS and NTS. Shown is the U40 spacer, which would be bound at the 3′ end of the gRNA. **c**, gRNA screen of Al3Cas12f across two loci in K562 cells using the 134-nt gRNA scaffold and a 20-nt spacer. Editing efficiency is represented as the mean percentage (*n* = 3 independent biological replicates) of indels detected in NGS reads.

Top-performing 27 gRNAs targeting the AAVS1 locus (dark blue) and the *APOA1* locus (light blue). **d**, Comparison of mammalian cell activity among Al3Cas12f, OsCas12f and LbCas12a across eight different loci. Editing efficiency is represented as the mean percentage of indels detected in NGS reads (*n* = 3 independent biological replicates). **e**, Box-and-whisker plot showing the data in **d**. The box outlines denote the first and third quartiles, the median is represented by the center line and the whiskers encompass the full data range. Individual data points are shown as dots. Significance was determined using a paired *t*-test.

R117 and R119 interact with G49 and G50, respectively, while D77 forms a hydrogen bond with the C57 (Fig. 3c and Extended Data Fig. 7). These observations reveal that the same residues in each RhCas12f monomer facilitate dual recognition of the PAM and gRNA, respectively, and prevent nonproductive PAM binding by the wrong monomer.

To examine whether the distal end of stem 2 is necessary for Cas12f activity, we deleted this region in the gRNA of OsCas12f (Δ40–58 nt) and RhCas12f (Δ50–60 nt) and assessed their activity in a plasmid interference assay. Deletion of the flexible region in stem 2 did not affect OsCas12f activity. In contrast, RhCas12f activity was significantly reduced (Fig. 3d), indicating that the interaction between Cas12f.2 and stem 2 of the gRNA is essential for RhCas12f activity. Therefore, engineering of stem 2 in Cas12f gRNAs could provide a strategy toward creating scaffolds to prevent the formation of nonproductive conformations of these enzymes.

The Al3Cas12f gRNA adopts a more compact structure relative to OsCas12f and RhCas12f. Al3Cas12f gRNA does not contain a stem 4 region, while its stem 2 adopts a kinked conformation that allows stem 2 to dock into the REC lobe of the protein (Fig. 3b). As our structures indicated that stem 4 in OsCas12f and RhCas12f did not form direct contacts with the protein, we speculated that deletion of this region could enhance the efficiency of these enzymes. Indeed, deletion of the stem 4 region (Δ106–125 nt) in OsCas12f showed a tenfold increase in plasmid interference (Fig. 3d). We conclude that Al3Cas12f contains a naturally optimized gRNA scaffold and supports previous findings that removal of stem 4 in Cas12f nucleases similarly enhances nuclease activity[23]. Overall, our results provide new mechanistic and

structural insights into the gRNA configuration and optimization of Cas12f nucleases.

## Variability in the Cas12f dimerization interface explains enhanced activity by Al3Cas12f

Cas12f dimerizes through interactions between the REC domains of its two monomers. OsCas12f and RhCas12f primarily form dimers through two-layer interactions; the top layer involves the first α-helix of the REC domain (α1), while the bottom layer is formed by two loops located below α1 (Fig. 4a–f). In contrast, the dimerization interface of Al3Cas12f is more complex compared to OsCas12f and RhCas12f (Fig. 4g–k). The REC domain of Al3Cas12f is larger, containing an additional 50–60 residues that form two α-helices (α2 and α3). These extended helices in REC.1 and REC.2 interact with the NTS DNA and the hairpin in stem 2 of the gRNA, respectively (Extended Data Fig. 8a–c). These REC domain interactions with nontarget DNA and gRNA have not been observed for other Cas12f nucleases and may result in enhanced stability of the complex.

Al3Cas12f contains an α-helix α1 in REC.2 (α1.2) that curves toward REC.1, while α1 of REC.1 (α1.1) remains mostly linear (Fig. 4g). The bending of α1.2 toward α1.1 allows α1.1 to dock onto the α1.2 helix, aided by multiple π–π stacking interactions. This unique dimerization mechanism is analogous to a mortise-and-tenon model, where the mortise is the concave slot of the receiving part (α1.2) and the tenon is the protruding structure that fits into the mortise (α1.1) (Fig. 4g). Moreover, Al3Cas12f contains two other mortise–tenon joints located alongside the helices of the REC domain (Fig. 4h,k). The stability at the interface

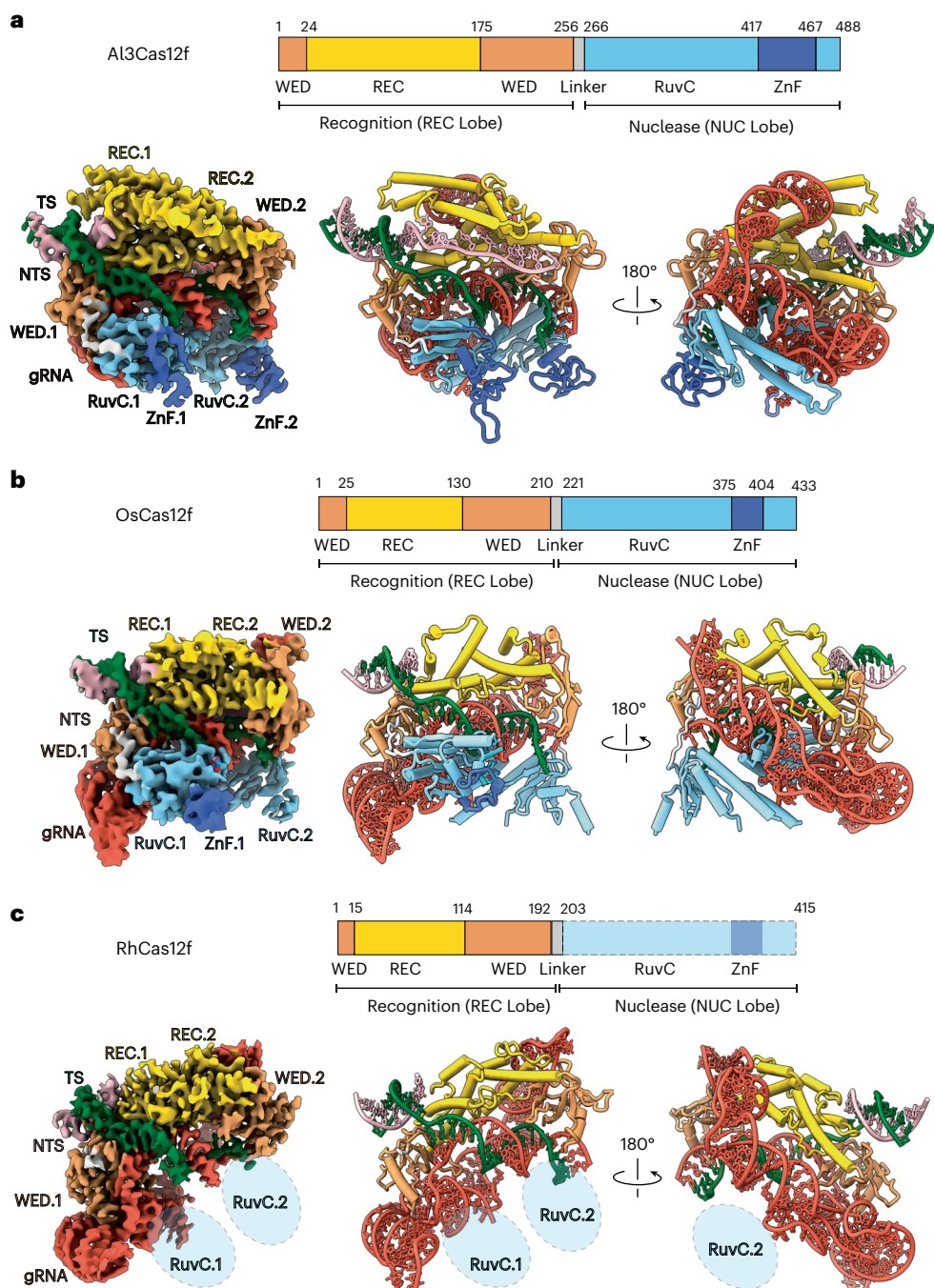

**Fig. 2 | Cryo-EM structure of Cas12f orthologs reveal distinct dimerization interface and RuvC plasticity. a–c**, Domain organization, cryo-EM maps and models for Al3Cas12f (**a**), OsCas12f (**b**) and RhCas12f (**c**). Density is colored on the basis of proximity to the modeled domains. gRNA is colored red, REC domains are colored gold, WED domains are colored sandy brown, RuvC domains are colored sky blue and ZnF domains are colored royal blue. The DNA TS and NTS are colored green and pink, respectively.

of Al3Cas12f is mediated through a combination of π–π stacking interactions between aromatic residues and van der Waals interactions among nonpolar amino acids (Fig. 4j). Additionally, hydrogen bonds and salt-bridge interactions have a notable role in the stabilization of the dimer (Fig. 4h,i,k).

To validate these observations, we performed alanine substitutions of residues involved in stabilizing the dimerization interface or directly contacting the gRNA or R-loop. These substitutions significantly reduced Al3Cas12f activity in *Escherichia coli* (Fig. 4l and Extended Data Fig. 8d). To further evaluate the impact of these substitutions, we examined how they influenced the stability of

the apo Cas12f dimer using mass photometry. Our results revealed that disrupting the dimer interface markedly compromises dimer stability (Extended Data Fig. 9). We then used covalently linked dimers to assess how individual amino acid changes affect the GFP depletion activity of Al3Cas12f. Notably, the F112A, R63A, Y60A and R49A substitutions, which disrupt key contacts within the dimer interface, led to a pronounced decrease in GFP depletion activity compared to wild type (Extended Data Fig. 9b). Meanwhile, the F102A substitution in monomer 1, which disrupts an important interaction with the NTS, exhibited significantly reduced activity in GFP depletion compared to the wild type. In contrast,

**Table 1 | Cryo-EM data collection, refinement and validation statistics**

| | Al3Cas12f state I | Al3Cas12f state II | OsCas12f state I | OsCas12f state II | OsCas12f state III | RhCas12f |
|---|---|---|---|---|---|---|
| **Data collection and processing** | | | | | | |
| Magnification | 150,000 | 150,000 | 150,000 | 150,000 | 150,000 | 105,000 |
| Voltage (kV) | 200 | 200 | 200 | 200 | 200 | 300 |
| Electron exposure (e⁻ per Å²) | 50 | 50 | 50 | 50 | 50 | 80 |
| Defocus range (μm) | −1.5 to −2.5 | −1.5 to −2.5 | −1.5 to −2.5 | −1.5 to −2.5 | −1.5 to −2.5 | −1.5 to −2.5 |
| Pixel size (Å) | 0.94 | 0.94 | 0.94 | 0.94 | 0.94 | 0.83 |
| Initial particle (no.) | 2,039,247 | 2,039,247 | 223,671 | 223,671 | 223,671 | 875,624 |
| Final particle (no.) | 141,537 | 106,204 | 132,962 | 44,815 | 42,153 | 316,144 |
| Map resolution (Å) | 3.17 | 3.20 | 3.27 | 3.67 | 3.68 | 3.07 |
| FSC threshold | 0.143 | 0.143 | 0.143 | 0.143 | 0.143 | 0.143 |
| **Refinement** | | | | | | |
| Initial model used | ModelAngelo | ModelAngelo | AlphaFold | 9NZT | 9NZT | AlphaFold |
| Model resolution (Å) | 3.19 | 3.25 | 3.2 | 3.7 | 3.6 | 3.1 |
| FSC threshold | 0.143 | 0.143 | 0.143 | 0.143 | 0.143 | 0.143 |
| Map sharpening $B$ factor (Å²) | −109.1 | −117.0 | −127.7 | −107.4 | −104.3 | −135.7 |
| Model composition | | | | | | |
| Nonhydrogen atoms | 9,025 | 11,066 | 7,928 | 9,260 | 9,448 | 7,397 |
| Protein residues | 694 | 889 | 598 | 750 | 728 | 389 |
| Nucleotides | 169 | 185 | 146 | 152 | 170 | 199 |
| Ligands | N/A | N/A | Zn: 1 | Zn: 1 | Zn: 1 | N/A |
| Mean $B$ factors (Å²) | | | | | | |
| Protein | 93.36 | 83.04 | 70.81 | 39.83 | 38.94 | 59.33 |
| Nucleotides | 164.30 | 117.10 | 94.92 | 64.73 | 67.14 | 151.79 |
| Ligands | N/A | N/A | 159.75 | 93.81 | 101.67 | N/A |
| Root-mean-square deviations | | | | | | |
| Bond lengths (Å) | 0.005 | 0.004 | 0.005 | 0.006 | 0.007 | 0.006 |
| Bond angles (°) | 0.883 | 0.843 | 1.035 | 1.216 | 1.250 | 1.116 |
| **Validation** | | | | | | |
| MolProbity score | 1.54 | 1.32 | 1.34 | 1.53 | 1.50 | 1.35 |
| Clashscore | 5.46 | 5.86 | 4.19 | 5.79 | 7.06 | 6.39 |
| Poor rotamers (%) | 0.61 | 0.53 | 0.77 | 1.09 | 0.48 | 0.28 |
| Ramachandran plot | | | | | | |
| Favored (%) | 97.38 | 98.40 | 97.29 | 96.88 | 97.47 | 98.70 |
| Allowed (%) | 2.62 | 1.60 | 2.71 | 3.12 | 2.53 | 1.30 |
| Disallowed (%) | 0 | 0 | 0 | 0 | 0 | 0 |
| Protein Data Bank | 9NZO | 9NZR | 9NZT | 9NZQ | 9NZS | 9NZP |
| EM Data Bank | EMD-49954 | EMD-49957 | EMD-49959 | EMD-49956 | EMD-49958 | EMD-49955 |

the same substitution in monomer 2, which abolishes an interaction with the gRNA, caused only a modest reduction in activity (Extended Data Fig. 9b).

To assess whether the REC domain extensions of Al3Cas12f are lineage specific or broadly conserved, we generated a Cas12f-focused multiple-sequence alignment (MSA) and phylogenetic tree using representative clades[27,28] (Supplementary Data 1). The alignment revealed a conserved region unique to the Al3Cas12f clade, whereas other clades showed substantial variability and differences in REC domain length, suggesting lineage-specific features (Supplementary Fig. 4). Because sequence comparisons may miss structural conservation, we aligned representative structures to the Al3Cas12f cryo-EM model, which showed strong REC domain structural similarity within the Al3Cas12f clade but divergence in other clades[21,22,26] (Supplementary Fig. 5).

Overall, the defining difference is the presence of extended α2 and α3 helices that engage the NTS and gRNA, supporting the idea that these REC domain features represent an adaptation unique to the Al3Cas12f lineage. Taken together, these findings demonstrate that Al3Cas12f has unique structural features that contribute to enhanced nuclease activity.

## Cas12f intermediate structures reveal differing activation mechanisms for Cas12f variants

To further investigate the molecular mechanisms underlying Cas12f activation, we analyzed our structural datasets for heterogeneity. While the RhCas12f dataset exhibited high structural homogeneity among the selected particles, multiple conformational states were resolved for both OsCas12f and Al3Cas12f.

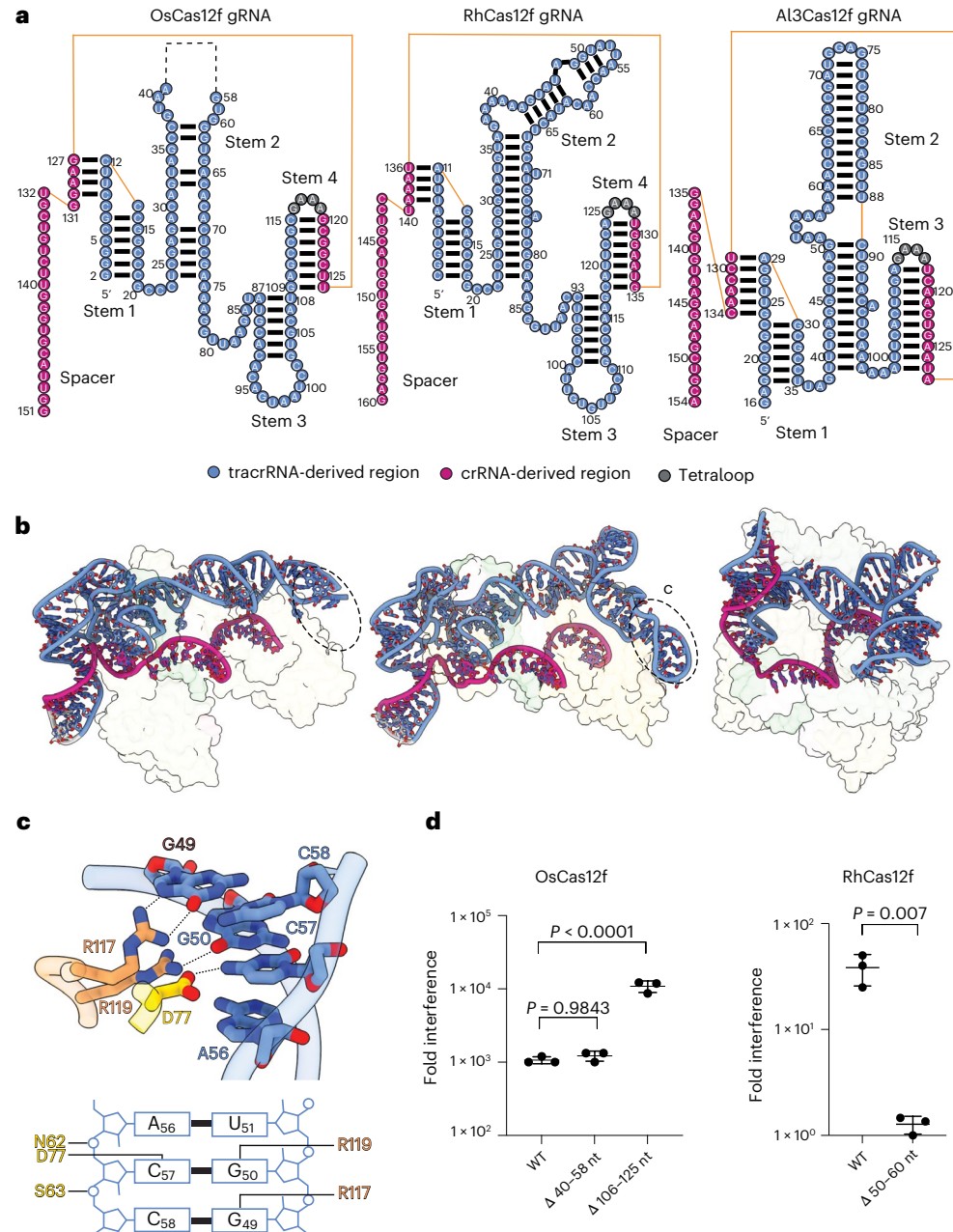

**Fig. 3 | Overview of gRNA architectures of OsCas12f, RhCas12f and Al3Cas12f.**
**a**, Schematic diagram of gRNA scaffolds of OsCas12f, RhCas12f and Al3Cas12f. The tracrRNA, crRNA and tetraloop are labeled and colored accordingly. Unresolved regions are indicated by a dashed line. **b**, Surface views of OsCas12f, RhCas12f and Al3Cas12f bound to gRNA. Dotted circles highlight the differences of stem 2 in OsCas12f and RhCas12f gRNA. This region is present in RhCas12f gRNA because of the PAM-analogous contacts highlighted in **c**. **c**, Zoomed-in overview of interactions between the RhCas12f WED/REC domains and stem 2 of its cognate gRNA. **d**, Plasmid interference activity of wild-type and gRNA truncation mutants in OsCas12f and RhCas12f. Data represent the mean ± s.d. (*n* = 3 independent biological replicates). Significance was determined using a one-way analysis of variance (ANOVA) for OsCas12f or unpaired *t*-test for RhCas12f.

We solved three distinct structures of OsCas12f in multiple R-loop formation and conformational states (Fig. 5a–c). State I showed an 8-bp R-loop with an unresolved RuvC.2, indicating this domain can sample multiple conformational states, as previously seen for CnCas12f (Fig. 5a). State II revealed the RuvC.2 domain docked onto the spacer region of the gRNA before R-loop completion, allowing us to model six additional nucleotides of the gRNA spacer alongside the 8-bp heteroduplex, while RuvC.2 was only partially resolved (Fig. 5b). In state III, the structure displayed a fully formed 20-bp R-loop with RuvC.2 domain partially resolved (Fig. 5c). Al3Cas12f displayed two different intermediates, both with a fully formed 20-bp R-loop: a preproduct state before RuvC.2 docking (state I) and a postproduct state with a fully docked RuvC.2 domain at the PAM-distal end of the spacer (state II) (Supplementary Fig. 6).

Although we show that the RuvC.1 domain in OsCas12f is nuclease active and cleaves both strands (Supplementary Fig. 7), our structures indicate that RuvC.2 undergoes notable structural rearrangements before eventually docking and stabilizing the PAM-distal region of the R-loop. This enables full R-loop formation in OsCas12f in state II. Conversely, Al3Cas12f can form a full R-loop before RuvC.2 docking. This discrepancy in the order of RuvC docking between OsCas12f and Al3Cas12f suggests mechanistic differences between these nucleases that may affect their cleavage efficiency.

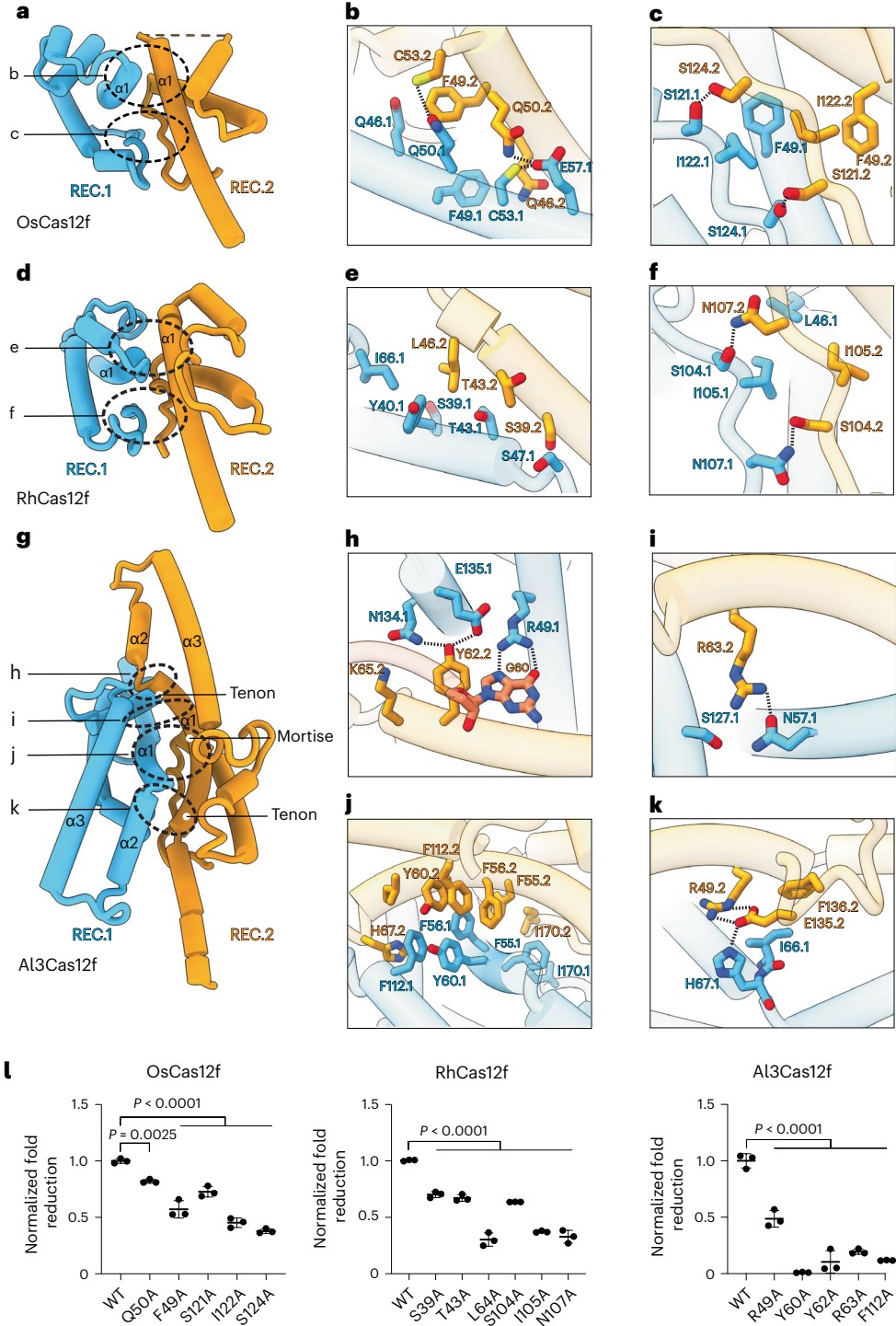

**Fig. 4 | REC dimerization interfaces in OsCas12f, RhCas12f and Al3Cas12f.**
**a**, Model of OsCas12f dimerization interface. **b,c**, Zoomed-in overview of specific
interactions that mediate the interactions among the REC interface in OsCas12f
in α1 (**b**) and loop (**c**). **d**, Model of RhCas12f dimerization interface. **e,f**, Zoomed-in
overview of specific interactions that mediate the interactions among the REC
interface in RhCas12f in α1 (**e**) and loop (**f**). **g**, Model of Al3Cas12f dimerization
interface. **h–k**, Zoomed-in overview of specific interactions that mediate the
interactions among the REC interface in Al3Cas12f in the tenon (**h**) and REC.1 α3
(**i**), α1 (**j**) and α2 (**k**). **l**, GFP assays evaluating the effect of different dimerization
domain mutants to disrupt specific interactions. Data represent the mean ± s.d.
(*n* = 3 independent biological replicates). Significance was determined using a
one-way ANOVA.

To understand the importance of these intermediate structures,
we examined the RuvC lid in OsCas12f and Al3Cas12f. States I and II
in OsCas12f (8-bp R-loop formed) show the RuvC lid in a 'closed' con-
formation (Fig. 5d). This loop region, connecting the fourth β strand
(β4) and the third α-helix (α3) of the RuvC domain, blocks access of
the nucleic acid to the catalytic active site and consequently serves as

a crucial regulator of dsDNA *cis*-cleavage. Upon target loading, this lid
has been shown to rearrange to allow for the DNA TS to pass through
to the active site[18,29–31]. The fully formed R-loop structure in OsCas12f
shows that the RuvC.1 lid is in an 'open' conformation, indicating that
the loop opens only after the R-loop is fully formed. Instead of forming
a helix as shown in Un1Cas12f (ref. 18), the middle region of the lid in

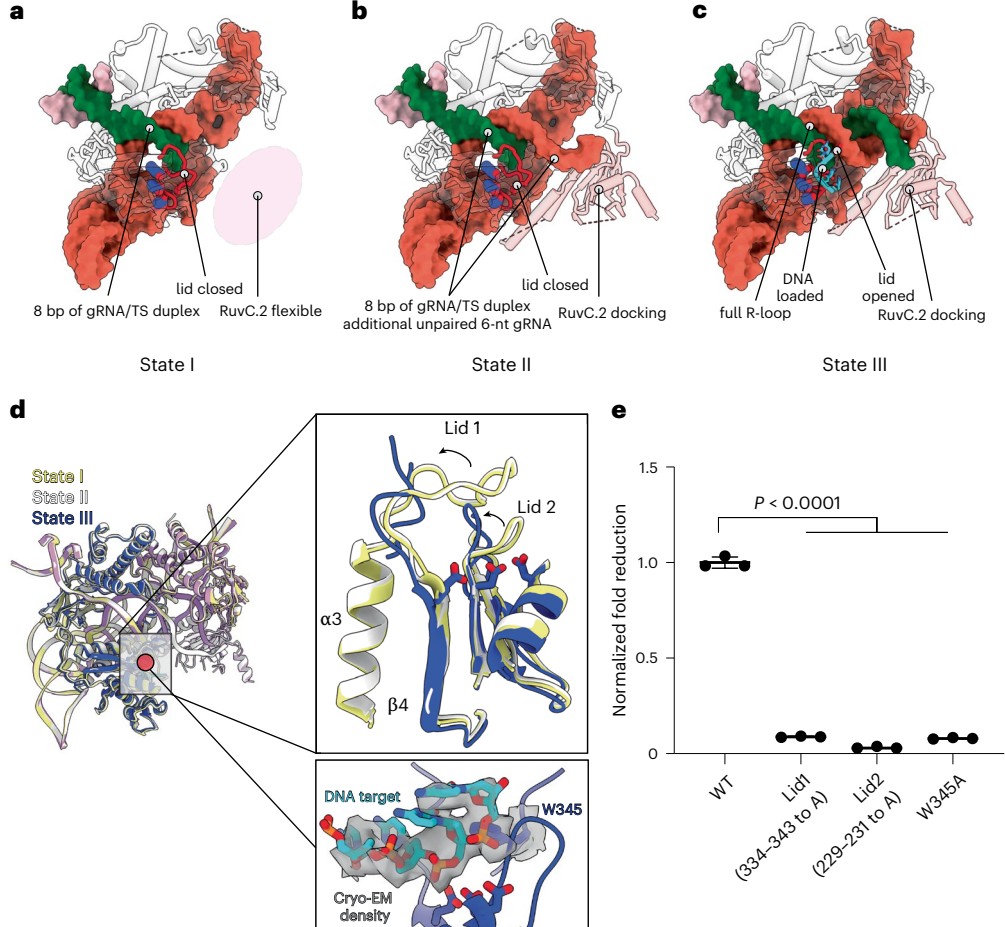

**Fig. 5 | DNA target loading is facilitated by a two-lid opening mechanism.**
**a**–**c**, Intermediate state structures illustrating conformational changes at three activation stages of OsCas12f: state I (**a**), state II (**b**) and state III (**c**). Active-site residues are shown in blue; the lid regions are shown in red. **d**, Structural alignment of intermediate state structures. The regions in the box and red circle are enlarged to illustrate the open-to-close conformational change of two lid regions (black arrow) and the position of the DNA substrate in the catalytic pocket. **e**, GFP depletion assays for lid mutations in OsCas12f. Data represent the mean ± SD ($n = 3$ independent biological replicates). Significance was determined using a one-way ANOVA.

OsCas12f State III structure is flexible and not modeled (D335–L342). We show that this lid is essential for cleavage by substituting E334–R343 to alanine, preventing the lid from undergoing the rearrangements necessary for cleavage and resulting in significantly reduced OsCas12f activity in GFP depletion assays (Fig. 5e). In addition to the RuvC loop, we noticed that W345 in OsCas12f positions DNA in the active site of the RuvC domain, like what has been observed in UnCas12f. A W345A substitution significantly hindered its activity in *E. coli*, indicating that this residue is essential for activity in OsCas12f (Fig. 5d,e). Moreover, we also observed a smaller loop region (residues 229–235, denoted as lid 2) that undergoes similar conformational changes. Alanine substitutions of the residues 229–231 abolished OsCas12f activity, indicating an essential role in nuclease activity (Fig. 5d,e).

In contrast to OsCas12f, both structures of Al3Cas12f showed the lid in an open conformation, indicating that Al3Cas12f can adapt its cleavage-active conformation before RuvC.2 docking. Overall, our structures reveal that RuvC.2 docking is essential for full R-loop formation for OsCas12f but not Al3Cas12f. As noted earlier, our results suggest that Cas12f nucleases have distinct cleavage mechanisms, modulated by RuvC rearrangements.

### Kinetic analysis of Cas12f variants

To further investigate the functional differences between these Cas12f variants, we used in vitro cleavage assays and stopped-flow fluorescence measurements of R-loop formation for a detailed kinetic analysis.

First, we measured cleavage of TS and NTS by mixing guide RNP with fluorescently labeled DNA to initiate the reaction. While all enzymes were slower than SpCas9, the cleavage rates varied widely among the Cas12f variants. Al3Cas12f had the fastest cleavage rate and both TS and NTS cleavage went to completion, with data that approximate a single exponential function (Extended Data Fig. 10). Both OsCas12f and RhCas12f displayed highly biphasic behavior with relatively low amplitudes of cleavage, followed by a slow linear approach to completion. This was the most pronounced for RhCas12f and prevented simple interpretation of the results from equation-based fitting.

Next, we measured R-loop formation using a stopped-flow fluorescence assay that we previously developed for SpCas9 (refs. 32,33). Data for each variant showed biphasic behavior, indicative of a more complex model than simple DNA binding followed by R-loop formation.

To resolve the complexities in both sets of experiments in the context of a unifying model, we fitted the data for each variant globally with KinTek Explorer simulation software[34,35]. Starting concentrations for each experiment and output observables were entered into the software, allowing us to test multiple models for Cas12f catalysis. Our model has a few unique features for these variants relative to previous models we published for SpCas9 to account for the complexities in the data. A nonproductive state before R-loop formation (EDx) was added in addition to the productive fully formed R-loop state (EDR) that is accessible from the enzyme–DNA (ED) state to account for the biphasic R-loop formation rate (Extended Data Fig. 10). We observed a

similar state for Cas9d, which was required to account for the biphasic R-loop formation data[36]. The data for the three variants showed a wide range of forward rate constants (0.05–4.5 s$^{-1}$) and reverse rate constants (0.001–2.14 s$^{-1}$). In particular, RhCas12f exits this state quite slowly, giving rise to the slow second phase of the cleavage kinetics for TS cleavage. A nonproductive state after NTS cleavage (XDP1) was included that is accessible from the cleaved NTS state (EDP1). This was required to account for the highly biphasic cleavage data for OsCas12f and RhCas12f. While this step was required to fit the data, only an equilibrium constant for this step could be obtained, as the rates of formation and decay of this state had to be faster than the subsequent cleavage step to properly fit the data (>0.2 s$^{-1}$). Our intermediate structures of OsCas12f reveal that this enzyme requires the RuvC to dock before R-loop completion. Consequently, this EDx state could be attributed to a partially formed R-loop before RuvC docking.

From this analysis, we can also quantify the preference for each enzyme to cleave TS versus NTS using flux analysis on the partitioning of the R-loop formed state (EDH). For Al3Cas12f and OsCas12f, 65% and 70% of the fully formed R-loop state partitioned toward cleavage of the NTS first, respectively. For RhCas12f, approximately 45% partitioned toward NTS cleavage, essentially showing little preference toward which strand is cleaved first. The unique feature of this class of enzymes, namely the singular active nuclease domain, likely explains these new kinetic observations. Structurally, the RuvC.1 domain remains highly flexible in RhCas12f, suggesting that it may rearrange more freely to cleave both strands without notable specificity toward one or the other. In previous studies, we performed magnesium-initiated cleavage experiments where the enzyme, gRNA and DNA are preincubated in the absence of magnesium and then magnesium is added to initiate the reaction[36,37]. For these enzymes, the measured rates of cleavage were slower than the measured rates of R-loop formation, indicating that the cleavage step was only partially rate limited by R-loop formation; thus, we were able to extract all rate constants without the need for the magnesium-initiated cleavage experiments.

Global data fitting also resolved rates of R-loop formation for each variant. Forward rates of R-loop formation ranged from 0.04 s$^{-1}$ for RhCas12f to 0.64 s$^{-1}$ for OsCas12f and reverse rates ranged from 0.08 s$^{-1}$ for Al3Cas12f and RhCas12f to 0.49 s$^{-1}$ for OsCas12f. More importantly, the equilibrium constant for R-loop formation ranged from 0.5 for RhCas12f to 2 for Al3Cas12f. For wild-type SpCas9, the equilibrium constant for R-loop formation was measured at 2 (ref. 38), indicating that R-loop formation for these Cas12f variants is either more reversible or similar to SpCas9 (Extended Data Fig. 10 and Supplementary Fig. 8a–c).

The higher efficiency and fidelity of these variants can be explained by the in vitro kinetics. The reversible R-loop step, coupled to the slower cleavage rates, reflects the general principles seen with higher-fidelity SpCas9 variants[38,39]. That is, with slow cleavage and reversible R-loop formation, off-target DNA has a chance to dissociate before cleavage. Al3Cas12f is the most efficient at cleaving its target, as it does not get stuck in the EDx or XDP1 state like the other two variants, likely because of full R-loop formation before RuvC.2 docking based on our structural studies.

### Structure-guided engineering of Al3Cas12f increases editing efficiency

Despite saturating levels of editing efficiency at AAVS1, wild-type Al3Cas12f showed <61% editing at other loci (for example, *SOD1*; Extended Data Fig. 3). To improve the activity of Al3Cas12f, enzyme variants were designed on the basis of the cryo-EM structure in combination with phylogenetic sequence analysis to determine functionally conserved residues. This approach focused on increasing the positive charge of residues within hydrogen-bonding distance to the DNA substrate and/or the gRNA. Most of these residues were in the WED domain (Supplementary Fig. 9). Because Al3Cas12f exists as an asymmetric homodimer, special care was taken to ensure that the intended mutation would not disrupt (1) residues critical

for the dimerization of the two protomers or (2) key protein–nucleic acid interactions necessary for binding.

We started by testing single substitutions and evaluating the editing efficiency of these variants. Six single-substitution variants with editing efficiency over 60% were chosen to create combinatorial mutations (Supplementary Fig. 10a). We tested 35 double-substitution and triple-substitution variants in K562 cells targeting one AAVS1 site and the *SOD1* sites, as well as one hexamutant that combined six substitutions (Supplementary Fig. 10b and Fig. 6a). In total, 21 of the 35 combinatorial mutants showed at least 75% editing efficiency at *SOD1* with the A9 gRNA with one quarter of the dose used in the previously described spacer length assay, 125 ng of mRNA and 50 pmol of gRNA (Fig. 6b). Only the hexamutant displayed lower levels of editing efficiency (~10%) than the wild type (~40%). From this assay the K79R;M190K;E222K variant (no. 53) was chosen as the lead Al3Cas12f variant, hereby referred to as Al3Cas12f RKK (Fig. 6a,b and Supplementary Data 1). Furthermore, about half of the protein variants led to at least 90% editing efficiency at the AAVS1 locus (Supplementary Fig. 10b). Al3Cas12f RKK was benchmarked against the wild-type enzyme using eight reference guides previously used for AsCas12f (ref. 21). Six of these guides displayed >80% indels, with up to a 26-fold increase in activity at the *EMC6* locus (Fig. 6c). Overall, these results show that Al3Cas12f RKK is a highly efficient and compact nuclease that could be further optimized for genome editing.

### Discussion

Cas12f nucleases are highly desirable because of their size and levels of genome-editing activity. Through mining of a large, assembly-driven metagenomics database, we identified a natural ortholog of Cas12f, Al3Cas12f, that exhibits high levels of activity in vitro and in mammalian cells. We show that this enzyme is advantageous because of its recognition of a minimally restrictive PAM, naturally optimized gRNA and ability to remain a highly stable dimer.

To better understand how Al3Cas12f achieves higher levels of editing relative to other Cas12f orthologs, we performed a comprehensive structural comparison across various members of this CRISPR subtype. Before this study, structures of various Cas12f nucleases were described[18–21,23,26]. Here, we solved cryo-EM structures of Al3Cas12f and two additional Cas12f nucleases, also reported to generate high levels of indels in mammalian cells. Structures of Al3Cas12f, OsCas12f and RhCas12f in combination with AsCas12f, UnCas12f and CnCas12f, showed notable differences along their dimerization interfaces, which have a key role in stabilizing the enzyme, with Al3Cas12f having the largest number of interacting residues and interface area (Supplementary Fig. 11). Interestingly, Hino et al. demonstrated that AsCas12f activity could be greatly improved by dimerization interface stabilization[23]. They used unbiased deep mutational scanning, which revealed specific dimer residues that increase dimer contacts, bringing engineered AsCas12f closer to the native dimer interface observed in the highly active Al3Cas12f. Additionally, the dimerization interface of Al3Cas12f features an extensive contact area forming twice the number of contacts compared to OsCas12f and RhCas12f, mediated mainly by strong protein–protein interactions between its two large REC domains. The dimerization interface is further stabilized by contacts with its gRNA and NTS DNA, which forms a highly stable paired region with an optimal size to allow the hairpin to interact with the REC domain, a feature unique to Al3Cas12f. gRNA engineering efforts have shown that enhancing duplex stability by promoting more stable base pairing greatly enhances activity in cells. While we show the gRNA of Al3Cas12f has already been naturally optimized and contacts the REC domain for additional stability, structures of RhCas12f also revealed that the second hairpin of the gRNA binds the PAM-interacting residues of the nonproductive monomer, further stabilizing the RNP.

Structures of intermediate states determined in this work show new conformational states of Cas12f nucleases. Previous studies

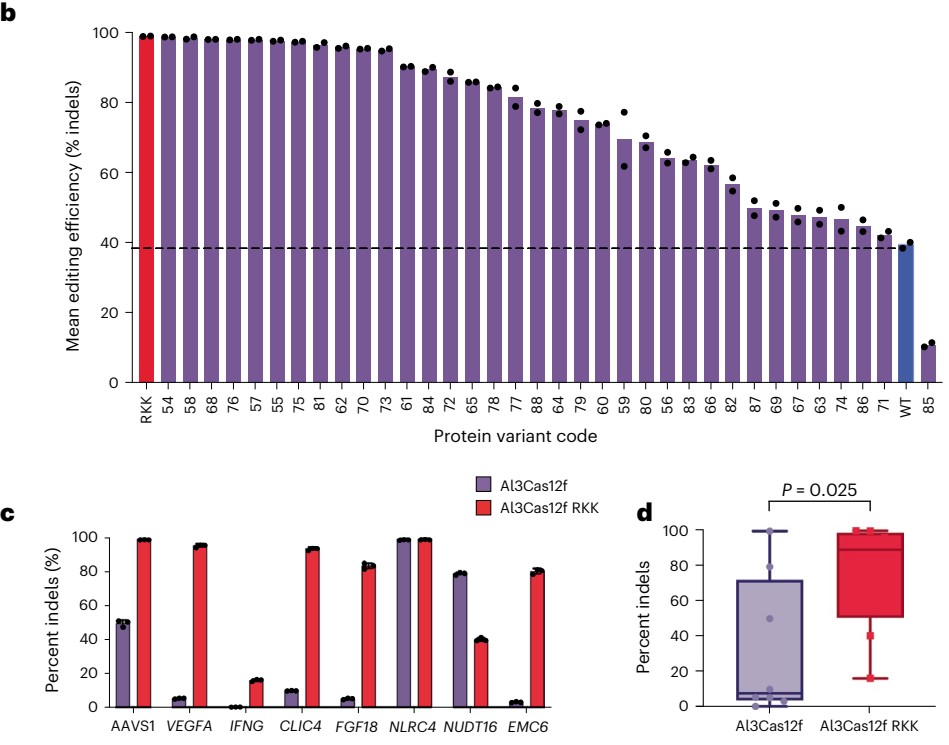

**a**, 

| Code | Variant | Code | Variant | Code | Variant | Code | Variant |
|---|---|---|---|---|---|---|---|
| RKK | K79R;M190K;E222K | 62 | E188K;E222K | 71 | E241K;N315E | 80 | M190K;E241K |
| 54 | M190K;E222K | 63 | E222K;E241K;N315E | 72 | K79R;M190K;E241K | 81 | E188K;M190K |
| 55 | M190K;E222K;N315E | 64 | M190K;E222K;E241K | 73 | M190K;N315E | 82 | K79R;E241K |
| 56 | E222K;E241K | 65 | K79R;E222K | 74 | M190K;E241K;N315E | 83 | E188K;E241K |
| 57 | K79R;E188K;E222K | 66 | E222K;N315E | 75 | K79R;M190K | 84 | K79R;E188K |
| 58 | E188K;M190K;E222K | 67 | K79R;E241K;N315E | 76 | K79R;E188K;M190K | 85 | K79R;E188K;M190K;E222K;E241K;N315E |
| 59 | K79R;E222K;E241K | 68 | K79R;M190K;N315E | 77 | K79R;E188K;N315E | 86 | K79R;N315E |
| 60 | K79R;E222K;N315E | 69 | E188K;E241K;N315E | 78 | K79R;E188K;E241K | 87 | E188K;M190K;E241K |
| 61 | E188K;E222K;N315E | 70 | E188K;M190K;N315E | 79 | E188K;N315E | 88 | E188K;E222K;E241K |

**Fig. 6 | Structure-guided engineering of Al3Cas12f furtherly improved editing efficiency. a**, Design of combinatorial mutants of Al3Cas12f. Engineered variants and codes are listed. **b**, Editing efficiency of 36 combinatorial mutants at *SOD1* in K562 cells in comparison to Al3Cas12f wild type. The bar height represents the mean percentage of indels (*n* = 2 independent biological replicates). Individual data points are shown as dots. The dashed line indicates the level of editing efficiency of Al3Cas12f wild type. The lead K79R;M190K;E222K variant (no. 53, in red) is referred to as RKK. **c**, Comparison of editing efficiency between Al3Cas12f wild type (purple) and RKK (red) across eight TTTA-PAM-containing genomic loci in K562 cells. Data represent the mean indel percentage ± s.d. (*n* = 3 independent biological replicates). **d**, Box-and-whisker plot showing the data in **c**. The box outlines denote the first and third quartiles, the median is represented by the center line and the whiskers encompass the full data range. Individual data points are shown as dots. Significance was determined using a paired *t*-test.

revealed that CRISPR–Cas systems exhibit extensive domain rearrangements sequentially with the formation of an R-loop[40]. Notably, intermediate structures of OsCas12f revealed a highly stable dimer throughout the entirety of R-loop formation. In contrast to other CRISPR–Cas systems, where REC domains are used for R-loop stabilization[40], OsCas12f uses its RuvC.2 domain to stabilize the PAM-distal end of the spacer and facilitate R-loop formation (Fig. 7a). However, Al3Cas12f uses extended REC domains to facilitate additional contacts with the R-loop and the NTS, enabling full R-loop formation before RuvC.2 docking (Fig. 7b). In contrast to both Al3Cas12f and OsCas12f, RhCas12f displays high flexibility on both RuvC domains, displaying a third, distinct mechanism of cleavage (Fig. 7c).

These structural observations are supported by our kinetics studies and models. We defined an EDx state, in which the enzyme is bound to its substrate but preceding completion of R-loop formation. Our kinetic analysis revealed that OsCas12f spends much longer in this state. This characteristic of OsCas12f can be explained by our 8-bp R-loop

intermediate structure, necessitating RuvC docking before adopting a catalytically active conformation. In contrast, both Al3Cas12f and RhCas12f do not transition favorably into this EDx state. Congruent with these observations, we were only capable of capturing structures of Al3Cas12f and RhCas12f with a fully formed R-loop. However, it must be mentioned that, while Al3Cas12f did show RuvC rearrangement to bind the PAM-distal end of the R-loop, this might not be necessary for RhCas12f as both RuvC domains were not defined in this structure. Our kinetic studies also revealed that R-loop formation is less reversible for Al3Cas12f, pulling the reaction toward the largely irreversible step of DNA cleavage.

Wild-type Al3Cas12f showed substantial locus-to-locus variability, raising the likelihood of insufficient editing at a given disease-relevant target. To address this potency limitation, we performed structure-guided engineering and generate an engineered variant, Al3Cas12f RKK, that increased average editing from <10% to >80% (under the tested conditions). This broad improvement

 765

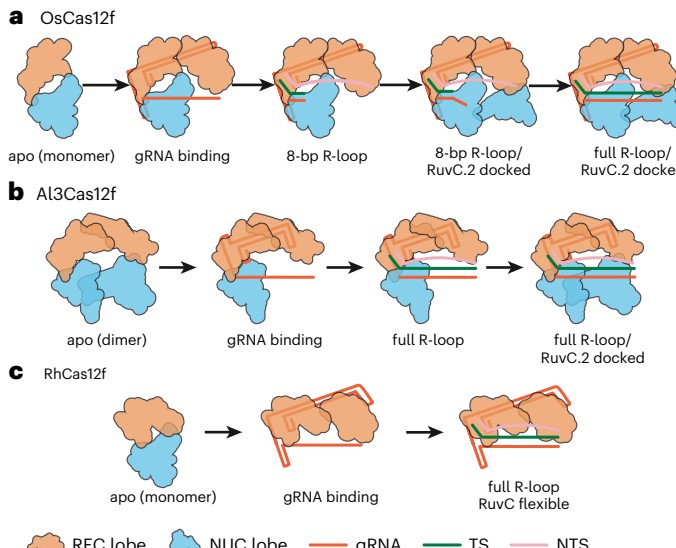

**a** OsCas12f

apo (monomer) → gRNA binding → 8-bp R-loop → 8-bp R-loop/RuvC.2 docked → full R-loop/RuvC.2 docked

**b** Al3Cas12f

apo (dimer) → gRNA binding → full R-loop → full R-loop/RuvC.2 docked

**c** RhCas12f

apo (monomer) → gRNA binding → full R-loop RuvC flexible

REC lobe   NUC lobe   gRNA   TS   NTS

**Fig. 7 | Activation mechanism for three Cas12f orthologs. a**, OsCas12f. Yhe apo protein exits as a monomer, and gRNA binding triggers dimer formation. Docking of RuvC.2 domain is required for full R-loop formation. **b**, Al3Cas12f. The apo protein exits as a dimer and full R-loop formation occurs before RuvC.2 domain docking. **c**, RhCas12f. The apo protein exits as a monomer and gRNA binding triggers dimer formation. Both RuvC domains remain highly flexible during R-loop formation.

across loci helps to overcome an activity threshold needed for development as a therapeutic gene-editing system. Consistent performance across multiple loci supports Al3Cas12f RKK as a more generally deployable platform and its high activity at lower input doses may be advantageous for improving the safety–efficacy balance in clinical translation.

Through a combination of structural, kinetic and biochemical assays, we highlighted multiple mechanistic differences among Cas12f orthologs that expand our understanding of the structural features important for achieving high nuclease activity. We showed that a highly stable dimerization interface in conjunction with a naturally optimized gRNA enables Al3Cas12f to form an efficient R-loop without requiring prior domain rearrangements. Lastly, we showed that an engineered variant increased gene-editing activity across multiple human genomic loci, making it well suited for a wide range of biotechnological and therapeutic applications that require compact nucleases for efficient delivery.

## Online content

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

Kaoling Guan [1,7], Rodrigo Fregoso Ocampo [1,2,7], Paula B. Matheus Carnevali[3], Cindy J. Castelle[3], Liliana Gonzalez-Osorio[3], Dominic T. Castanzo[3], Nicole C. Thomas[3], Molly Brothers[3], Tyler L. Dangerfield [1,6], Matthew M. Hooper [1,2], Madeline S. West[1], Nathan M. Appleby[1,2], Isabella Krudop[1], Rebecca C. Lamothe[3], Daniela S. Aliaga Goltsman[3], Lisa M. Alexander [3], Cristina N. Butterfield[3], Kenneth A. Johnson[1], Christopher T. Brown [3] & David W. Taylor [1,2,4,5] ✉

[1]Department of Molecular Biosciences, University of Texas at Austin, Austin, TX, USA. [2]Interdisciplinary Life Sciences Graduate Programs, University of Texas at Austin, Austin, TX, USA. [3]Metagenomi Therapeutics, Emeryville, CA, USA. [4]Center for Systems and Synthetic Biology, University of Texas at Austin, Austin, TX, USA. [5]LIVESTRONG Cancer Institutes, Dell Medical School, University of Texas at Austin, Austin, TX, USA. [6]Present address: Department of Chemistry and Biochemistry, University of Nevada, Las Vegas, NV, USA. [7]These authors contributed equally: Kaoling Guan, Rodrigo Fregoso Ocampo. ✉e-mail: dtaylor@utexas.edu

## Methods

### Sample collection and sequencing

Microbiome samples from abandoned animal stool were processed for metagenomic sequencing as described in Goltsman et al.[14]. Other microbiome sequencing data publicly available were downloaded from the National Center for Biotechnology Information Sequence Read Archive. A list of samples is provided in Supplementary Data 1. Raw sequencing reads were trimmed using BBMap (https://sourceforge.net/projects/bbmap/) and assembled using Megahit[41]; genes were predicted using Prodigal[42].

### Identification of compact type V nucleases

The discovery of compact type V nucleases was based on homology searches performed using the HMMER software. Type V nuclease sequence hits were retained if they met the following criteria: (1) the hmmsearch $e$ value was $\leq 10^{-5}$; (2) the genes encoding the nuclease were within 1 kb of a CRISPR array; and (3) the amino acid sequence length ranged between 350 and 700 aa. MMSeqs2 (https://github.com/soedinglab/MMseqs2) was used to cluster sequences at 100% amino acid identity, with coverage mode 1 and 80% coverage of the target sequence (parameters: --cov-mode 1 -c 0.8 --min-seq-id 1.0).

Sequence representatives and known reference sequences (Supplementary Data 1) were chosen to build an MSA using MAFFT (https://mafft.cbrc.jp/alignment/software/) with the Needleman–Wunsch algorithm for global alignment and FastTree (https://doi.org/10.1371/journal.pone.0009490) was used to build a phylogenetic tree in Geneious Prime 2025.2.2 (https://www.geneious.com). Examination of individual clades on the phylogenetic tree, including the nuclease gene's genomic context, led to the identification of novel Cas12f-M119 sequences.

The naming scheme used here is as follows: the type V system most closely related to these sequences (that is, Cas12f) is followed by the suffix MGX-Y, where MG indicates that the proteins are derived from metagenomic fragments, X represents the family identifier and Y indicates the member identifier. For example, Cas12f-MG119-2, a Cas12f enzyme recovered from metagenomics data, is the member 2 of family 119.

In the case of Cas12f-MG119-28, the consensus taxonomy of all the proteins encoded in the contig was used to assign a taxonomic affiliation with Kaiju (version 15)[43]. Most proteins encoded in the contig in which Cas12f-MG119-28 was identified were classified as belonging to *Alistipes* sp. 58_9_plus (strain Z76).

### Search for novel tracrRNA sequences and gRNA design

To identify putative tracrRNA sequences, the intergenic sequence and a minimal array adjacent to the Al3Cas12f nuclease gene were expressed in transcription–translation reaction mixtures using myTXTL Sigma 70 master mix kit (Arbor Biosciences). Cleavage activity was confirmed by mixing 5 nM of a target plasmid DNA library representing all possible 8N PAMs, a fivefold dilution of the TXTL expressions, 10 nM Tris-HCl, 10 nM MgCl₂ and 100 mM NaCl at 37 °C for 2 h. Cleavage was verified by next-generation sequencing (NGS) following previously reported protocols[44].

To obtain the sequence of the tracrRNA and the crRNA, RNA was extracted from TXTL lysate following the Quick-RNA miniprep kit (Zymo Research) and eluted in water. The total RNA from each sample were prepped for RNA sequencing using the New England Biolabs (NEB) Next small RNA library prep set for Illumina. Amplicons between 150 and 300 bp were quantified by TapeStation and Qubit and pooled to a final concentration of 4 nM. A final concentration of 12.5 pM was loaded into a MiSeq V3 kit and sequenced in a Miseq system (Illumina) for 176 total cycles. Sequencing reads were used to identify the tracrRNA sequence by mapping back to the original DNA fragments encoding the Cas12f-MG119-2 system.

To identify additional noncoding regions containing potential tracrRNAs, the sequence of the active Cas12f-MG119-2 tracrRNA was mapped to other DNA fragments encoding homologs (for example, Cas12f-MG119-1 and Cas12f-MG119-3). The 3′ end of the predicted tracrRNA sequences and the 5′ end of the repeat sequences were trimmed and then connected with a GAAA tetraloop to generate the gRNAs. The predicted gRNAs were subjected to a PAM determination assay. The verified sequences for Cas12f-MG119-2 and Cas12f-MG119-3 were used to generate covariance models to predict tracrRNAs for other homologous systems using MSA of the active and predicted tracrRNA sequences. The secondary structure of the MSA was obtained with RNAalifold (Vienna Package) and the covariance models were built with Infernal packages (http://eddylab.org/infernal/). The gRNAs were tested using in vitro cleavage reactions as described below and activity was confirmed for short (134 nt without spacer; Supplementary Data 1) and a long (152 nt without spacer; Supplementary Data 1 and Supplementary Fig. 1) versions of the gRNA.

### In vitro cleavage reactions to enable PAM determination

First, 5 nM of nuclease-amplified DNA templates and 25 nM of gRNA-amplified DNA templates, including one of the spacer sequences 5′-GTCGAGGCTTGCGACGTGGT-3′ (U67 spacer) or 5′-TGGAGATATCTTGAACCTTG-3′ (U40 spacer), were expressed at 37 °C for 3 h with a PURExpress in vitro protein synthesis kit (NEB). Plasmid library DNA cleavage reactions were carried out by mixing 5 nM of the target library representing all possible 8N PAMs, a fivefold dilution of PURExpress expressions, 10 mM Tris-HCl pH 7.9, 10 mM MgCl₂, 100 μg ml⁻¹ BSA and 50 mM NaCl (NEB 2.1 buffer) at 37 °C for 2 h. Reactions were stopped and cleaned with HighPrep PCR cleanup beads (MAGBIO Genomics) and eluted in Tris-EDTA pH 8.0 buffer.

To determine the PAM sequence and the TS cleavage site, 3 nM of the cleavage product ends were blunted with 3.33 μM dNTPs, 1× T4 DNA ligase buffer and 0.167 U per μl of Klenow fragment (NEB) at 25 °C for 15 min. To confirm the PAM sequence and determine the NTS cleavage site, 1.5 nM of the cleavage products were ligated with 150 nM adaptors, 1× T4 DNA ligase buffer (NEB) and 20 U per μl T4 DNA ligase (NEB) at room temperature for 20 min. The ligated products were amplified by PCR with NGS primers and sequenced by NGS to obtain the PAM sequence. Active systems that successfully cleaved the PAM library yielded a band around 188 or 205 bp in an agarose gel when blunted with Klenow fragment and a 195-bp band when blunted with mung bean nuclease (Supplementary Fig. 2).

### Nuclease mRNA production

To assess editing efficiency of wild-type Al3Cas12f in mammalian cells, nuclease mRNA was codon-optimized for human expression (GeneArt), then synthesized and cloned into a high-copy ampicillin plasmid (Twist Biosciences). Synthesized constructs were digested from the Twist backbone with HindII and BamHI (NEB) and ligated into a pUC19 plasmid backbone with T4 DNA ligase and 1× reaction buffer (NEB). Nuclease mRNA was synthesized by in vitro transcription (IVT) using the linearized nuclease mRNA plasmid. This plasmid was linearized by incubation of 50 μg of pDNA at 37 °C for 3 h with 50 U of SpeI-HF (NEB) enzyme in 1× reaction buffer. Linearized plasmid was purified with SureClean (Meridian), precipitated in ethanol and resuspended in nuclease-free water at an adjusted concentration of 1,000 ng μl⁻¹. The IVT reaction to generate nuclease mRNA was performed at 37 °C for 3 h using 1.5 μg of linearized plasmid; 5 mM ATP, CTP, GTP (all from NEB) and $N^{1}$-methyl pseudo-UTP (TriLink), HiScribe T7 RNA polymerase mix (NEB), 4 mM CleanCap AG (TriLink), 1× HiScribe transcription buffer (NEB) and 5 mM DTT. After 3 h, the plasmid DNA was digested with the addition of 250 U per ml DNase I (NEB) and incubated for 30 min at 37 °C. Purification of nuclease mRNA was performed using an RNEasy maxi prep kit (Qiagen) using the standard manufacturer protocol. Transcripts were further analyzed by capillary gel electrophoresis on a Fragment Analyzer (Agilent).

To study the performance of Al3Cas12f engineered variants in mammalian cells, sequences for the wild-type Al3Cas12f and the combinatorial mutant's mRNA were codon-optimized for human expression and synthesized as gBlocks (Twist Biosciences). DNA templates were used to generate nuclease mRNA at 50 °C for 1 h under similar conditions as outlined above but altering the following components: 2,000 ng of amplified DNA template, 2.5 mM DTT, 2.5 U per ml inorganic *E. coli* pyrophosphatase (NEB) and 1,000 U per ml murine RNase Inhibitor (NEB). Once the reaction was complete DNA was digested with the addition of 250 U per ml DNase I (NEB) and incubated for 10 min at 37 °C. Purification of nuclease mRNA was performed using an RNeasy maxi kit (Qiagen) using the standard manufacturer protocol. Transcript quality was analyzed by capillary gel electrophoresis on a Fragment Analyzer (Agilent). mRNA with ≥70% purity or single bands corresponding to full-length transcripts were used in the described assays.

### Editing efficiency in mammalian cells

Spacer sequences targeting human loci were identified by searching the albumin intron 1, *APOA1* exon 3 and AAVS1 loci for 'TTR' PAMs to reduce the number of gRNAs tested for an initial screen. gRNAs were designed by appending the adjacent 20-nt regions to the 152-nt gRNA scaffold. After an initial gRNA screen with a total of 104 gRNAs, 30 of them were selected for a second round of validation (Supplementary Data 1), appending the spacer sequences to the minimal 134-nt gRNA scaffold. The top 27 gRNAs (*n* = 3) targeting AAVS1 and *APOA1* are shown in Fig. 1c. For nucleofection with Al3Cas12f mRNA, 500 ng of mRNA and 200 pmol of gRNA were mixed and incubated on ice until cells were prepared.

Two spacers that previously showed <50% editing efficiency were chosen to assess the performance of the Al3Cas12f engineered variants. For nucleofection, protein mRNA, with either *SOD1* A9 gRNA or AAVS1 F4 gRNAs, 125 ng of enzyme mRNA and 50 pmol of gRNA (134-nt gRNA scaffold and 19-nt spacer; Supplementary Data 1) were mixed together and incubated on ice until cells were prepared. Experiments were performed in K562 cells grown and passaged in Iscove's modified Dulbecco's medium (Gibco) supplemented with 10% (v/v) FBS at 37 °C with 5% $CO_2$. Approximately $1 \times 10^5$ cells were transfected with mRNA plus gRNA as recommended by the Amaxa 4D-Nucleofector protocol in a 4D-Nucleofector system (Lonza). Transfected cells were grown for 3 days, harvested and gDNA was extracted with QuickExtract (Lucigen) per the manufacturer's instructions. Targeted regions for indels were amplified using Q5 high-fidelity DNA polymerase (NEB) with primers and extracted DNA as the templates. PCR products were purified by HighPrep PCR cleanup system (MAGBIO) per the manufacturer's instructions. PCR primers appropriate for use in NGS-based DNA sequencing were generated, optimized and used to amplify the individual target sequences for each gRNA. The amplicons were sequenced on an Illumina MiSeq machine and analyzed with CRISPResso2 (https://github.com/pinellolab/CRISPResso2) to measure indel frequency.

### Al3Cas12f editing efficiency in K562 cells with different spacer lengths

To assess whether a spacer length different from 20 nt (18, 19, 21, 22, 23, 24 and 25 nt) would have an effect on editing efficiency, we selected five gRNAs previously screened with the 20-nt-long spacer and a truncated version of the 134-nt gRNA scaffold. Three of these guides target the *SOD1* locus and displayed editing efficiencies between ~14% and 46 % in the past. The two other guides targeted the safe harbor locus AAVS1, previously used to benchmark activity (Fig. 1e). Editing efficiency in K562 cells was tested following the same methods described in the main text.

### Phylogenetic tree of Cas12f sequences

Representative Cas12f sequences from the literature (Supplementary Data 1), members of the Al3Cas12f clade and other Cas12f-MG sequences in this work (Supplementary Data 1) were aligned in

Geneious Prime 2025.2.2 (https://www.geneious.com) using MAFFT (Needleman–Wunsch global alignment) and a phylogenetic tree was built using FastTree (Gamma20 likelihood).

### Three-dimensional structure prediction and structural alignments

The amino acid sequence and corresponding gRNA sequence of Cas12f-MG119-1, Cas12f-MG119-2 and Cas12f-MG119-3 (Supplementary Data 1) and reference sequences Al1Cas12f1 and Al2Cas12f1 (ref. 27) (Supplementary Data 1) were used to predict their three-dimensional (3D) structure with Boltz2 (https://github.com/jwohlwend/boltz), a model for biomolecular interaction prediction[45]. For the purpose of this analysis, the structures were predicted as monomers interacting with their respective gRNA. Additionally, The cryo-EM structure of the CnCas12f1–gRNA–DNA complex (PDB 8HR5), the AsCas12f1–gRNAv1–dsDNA ternary complex (PDB 7WJU) and SpCas12f1 in complex with gRNA and cognate DNA (PDB 9I8Y) were obtained from the Protein Data Bank (https://www.rcsb.org/). The predicted and existing structures were visualized and individually aligned to the structure of Al3Cas12f. PyMOL (Schrödinger) was used to detect structural homology in a sequence-independent manner. To highlight the visual similarity among the REC domains, the second protomer and protein domains other than the REC domain in the first Al3Cas12f protomer (REC1) were hidden.

### Protein and gRNA production

Al3Cas12f proteins were expressed in the pMGBΔ vector. *E. coli* (NEBExpress Iq Competent *E. coli*; NEB, C3037I) cultures were grown at 37 °C in 2× YT medium (1.6% tryptone, 1% yeast extract and 0.5% NaCl) or Terrific Broth medium (Teknova T0690) with 100 µg $L^{-1}$ carbenicillin. At an optical density at 600 nm ($OD_{600}$) ≈ 0.8–1.2, cultures were induced with 0.5 mM IPTG (GoldBio I2481) and incubated at 18 °C overnight or 24 °C for 4–6 h, depending on construct. Cultures were then harvested by centrifugation at 6,000*g* for 10 min and pellets were resuspended in nickel A buffer (50 mM Tris-HCl pH 7.5, 750 mM NaCl, 10 mM $MgCl_2$, 20 mM imidazole, 0.5 mM EDTA, 5% glycerol and 0.5 mM TCEP), supplemented with protease inhibitors (Pierce Protease Inhibitor Tablets, EDTA-free; Thermo Fisher, A32965), and stored at −80 °C.

Cell pellets were thawed and resuspended in 120 ml with nickel A buffer + 0.5% *n*-octyl-β-D-glucoside detergent (P212121 and CI-00234). Samples were sonicated in an ice-water bath at 75% amplitude for a total processing time of 3 min using a 15 s on, 45 s off cycle. Lysates were clarified by centrifugation at 30,000*g* for 25 min and supernatants were batch-bound to 5 ml of Ni-NTA resin (HisPur Ni-NTA resin; Thermo Fisher, 88223) for ≥20 min.

Samples were loaded onto a gravity column, washed with 30 column volumes of nickel A buffer and then eluted in four column volumes of nickel B buffer (nickel A buffer + 250 mM imidazole) before concentrating in an Amicon Ultra-15 concentrator (50-kDa molecular weight cutoff; MilliporeSigma, UFC9050). Samples were taken throughout the purification process and run on an SDS–PAGE protein gel (Bio-Rad, 4568126), which was imaged on a Bio-Rad ChemiDoc in the stain-free channel following 5 min of ultraviolet activation (Extended Data Fig. 2). Protein samples were incubated with HRV-3C protease (GenScript, Z03092) for 15 min at room temperature, then loaded onto an S200i 10/300 GL column (Cytiva, 28-9909-44) and run in SEC buffer (50 mM Tris-HCl pH 7.5, 250 mM NaCl, 10 mM $MgCl_2$, 0.5 mM EDTA, 5% glycerol and 0.5 mM TCEP).

OsCas12f and RhCas12f proteins were produced and purified as previously described with minor modifications[17]. Briefly, protein samples were produced overnight in *E. coli* Nico21(DE3) (NEB) in the presence of 0.5 mM IPTG at 16 °C. Cells were harvested by centrifugation and lysed by sonication in lysis buffer (50 mM Tris-HCl pH 8.0, 50 mM imidazole and 1.5 M NaCl). The lysate was clarified by centrifugation at 38,000*g* for 45 min at 4 °C. The supernatant was loaded onto HisTrap

HP column (Cytiva) for affinity purification. Protein was eluted with an imidazole gradient buffer. The eluate was loaded onto HiTrap Heparin HP column (Cytiva) and eluted using a linear gradient of increasing NaCl concentration from 0.3 M to 2.0 M. The collected eluate was concentrated and loaded onto a Superdex 200 Increase 10/300 column (Cytiva) preequilibrated with SEC buffer (20 mM Tris-HCl pH 8.0, 500 mM NaCl, 0.5 mM TCEP and 5 mM MgCl$_2$).

For gRNA production, the linearized plasmid encoding gRNA sequence was used as a template for IVT using the HiScribe T7 high-yield RNA synthesis kit (NEB) following the manufacturer's instructions.

## Cryo-EM sample preparation, data collection and processing

Before complex assembly, gRNA was heated at 95 °C for 2 min, followed by cooling at room temperature, while the dsDNA duplex was formed by mixing equimolar TS and NTS, heating at 95 °C for 2 min and then cooling at room temperature. The Cas12f–gRNA binary complex was assembled by mixing purified Cas12f protein and gRNA at a molar ratio of 1:1.2 followed by incubation at 37 °C for 30 min. dsDNA duplex (the same amount as guide RNP) was then incubated with the binary complex at room temperature for 30 min to form the Cas12f–gRNA–DNA ternary complex. Next, 7 μM ternary complex was applied onto the Quantifoil R1.2/1.3 400-mesh copper grid that was plasma-cleaned for 1 min by a Solarus 950 plasma cleaner (Gatan). Grids were blotted by a Vitrobot Mark IV (Thermo Fisher) for 6 s with blot force of 0 at 4 °C and 100% humidity and plunge-frozen in liquid ethane. Grids were stored in liquid nitrogen before screening.

Grids were screened with SerialEM[46] on an FEI Glacios cryo-TEM (transmission EM) instrument. For Al3Cas12f and OsCas12f ternary complexes, images were collected on an FEI Glacios cryo-TEM instrument equipped with a Falcon 4 detector with a pixel size of 0.933 Å, while the images of the RhCas12f ternary complex were collected on an FEI Titan Krios cryo-TEM equipped with a Gatan K3 direct electron detector with a pixel size of 0.8332 Å. The defocus range was set to −1.5 to −2.5 μm. Motion correction, contrast transfer function (CTF) estimation and particle picking were carried out in cryoSPARC live (version 4.0). All subsequent data processing was carried out in cryoSPARC (version 4.4)[47].

For Al3Cas12f, 11,080 videos were collected and 2,039,247 particles were extracted for downstream analysis. Particles were classified using two-dimensional (2D) classification and classes with high resolution were picked for ab initio reconstruction. The resulting particles and volumes were subjected to one round of heterogeneous refinement. The best class was then refined through nonuniform refinement, yielding a 3.06-Å structure comprising 507,067 particles. To sort out the heterogeneity in the dataset, these particles were subjected to a single round of 3D classification. Classes with and without a visible RuvC domain were combined and particles were reextracted and refined to form the final structures (Extended Data Figs. 4 and 5).

For OsCas12f, 363,046 particles were picked while 223,961 particles were selected from 2D classification for further ab initio reconstruction (three classes). The resulting particles and volume classes were subjected to heterogeneous and nonuniform refinement, which yielded a subset of 132,962 particles and a volume at 3.45 Å. The particles were reextracted in a box size of 320 pixels followed by nonuniform refinement, as well as global and local CTF refinement, which improved volume resolution to 3.27 Å that was used for modeling. Particles were subjected to 3D classification, resulting in three structures (state I, state II and state III) with different R-loop formation states and RuvC.2 availability (Extended Data Figs. 4 and 5).

Data processing of the RhCas12f ternary complex was performed similarly to the method described for OsCas12f. A total of 875,624 particles were selected from 2D classification for ab initio reconstruction (three classes). The resulting particles and volume classes were subjected to heterogeneous and nonuniform refinement, which yielded a subset of 316,144 particles and a volume at 3.44 Å. The particles were reextracted in a box size of 320 pixels followed by nonuniform refinement, which yielded a 3.07-Å volume that was used for modeling (Extended Data Figs. 4 and 5).

## Model building and refinement

The Cas12f protein structure predicted by AlphaFold2 (ref. 48) was fitted into the ternary complex map as a rigid body in ChimeraX[49]. gRNA was modeled on the basis of secondary-structure prediction and the gRNA architecture of AsCas12f. The model was manually refined in Coot[50] and automatically refined by real_space_refine in PHENIX[51]. All structural figures were generated using ChimeraX (version 1.7.1). Al3Cas12f was built using Cosmic2 ModelAngelo[52,53]. The model was subsequently manually refined using Coot and Isolde[54] and automatically refined by real_space_refine in PHENIX.

## Mass photometry

Microscope coverslips (VMR) were assembled into flow chambers using double-sided tape. To build a calibration curve, protein stock controls with different molecular weights were measured in new flow chambers at a concentration of 20 nM. For all measurements, 15 μl of 20 nM effector diluted in 1× effector buffer was introduced into the flow chamber. After allowing the microscope to focus, 60-s videos were recorded. Consistent with Sonn-Segev et al.[55], mass photometry videos were recorded at 1 kHz, with exposure times varying between 0.6 and 0.9 ms, adjusted to maximize camera counts while avoiding saturation. All movies were analyzed using AcquireMP (Refeyn, 2024 R2).

## GFP depletion assay

The Cas12f coding sequence and DNA template of gRNA were cloned into *araBAD*-driven plasmids with resistance to chloramphenicol and carbenicillin, respectively. The GFP-encoding plasmid with streptomycin resistance was used as the target. Then, 5 μl of *E. coli* DH5α overnight culture bearing these three plasmids was subcultured with 2.5 ml of fresh Luria–Bertani (LB) medium. The culture was then split into two equal volumes: (1) as an uninduced control and (2) induced with 20 mM L-arabinose. All cultures were distributed into a 96-well plate (Invitrogen) and then incubated at 37 °C at 300 rpm. GFP fluorescence was monitored overnight by a CLARIOstar Plus plate reader (BMG Labtech). Each measurement was repeated three times. Fold reduction was calculated by comparing fluorescence between control and induced samples.

## In vivo plasmid interference assay

*E. coli* cells bearing three plasmids (Cas12f-encoding, gRNA-encoding and target plasmid) were grown to an OD$_{600}$ of approximately 0.6 at 37 °C. The culture was then split into two equal volumes: (1) as an uninduced control and (2) induced with 20 mM L-arabinose. All cell cultures were incubated at 16 °C overnight and subsequently serially diluted (six series of tenfold dilution) in fresh LB medium and spotted as 4-μl drops on LB agar plates containing all three antibiotics. LB agar plates were incubated at 37 °C overnight. Colonies were counted in the highest countable spot in the dilution series to calculate the number of colony-forming units per ml. The results were compared between control and induced sample to calculate the fold reduction because of nuclease-driven plasmid cleavage. Each experiment was repeated three times as biological replicates.

## DNA cleavage kinetics

To obtain the active fraction of the enzyme, an active-site titration assay was performed by incubating increasing concentrations of protein relative to a fixed concentration of DNA. Briefly, preassembled RNP was incubated at different ratios relative to a 10 nM DNA substrate in 10× effector buffer (100 mM NaCl, 10 mM Tris-HCl and 10 mM MgCl$_2$) and incubated at 37 °C or 46 °C for 60 min. For analysis, 2 μl of product was mixed with 10 μl of highly deionized (HiDi) formamide, then

resolved and quantified using an Applied Biosystems DNA sequencer (ABI 3130xl) equipped with a 36-cm capillary array and nanoPOP6 polymer. The percentage of cleaved DNA was evaluated by quantifying the ratio of the cleaved product relative to the uncleaved product for each concentration. All concentrations were plotted and the active fraction was determined by fitting a line through all data points using a standard $y = mx + b$ fit, where $b = 0$.

For time-course assays, reactions were initiated by adding 20 nM 6-FAM-labeled DNA substrate (for OsCas12f and RhCas12f and 10 nM for Al3Cas12f) into preassembled Cas12f–gRNA complex (100 nM active-site concentration of Cas12f), in 1× effector buffer, and then quenched at various times by mixing with equal volume of 0.5 M EDTA. Reactions were performed at 46 °C for OsCas12f and RhCas12f and at 37 °C for Al3Cas12f. Next, 2 μl of product was mixed with 10 μl of HiDi formamide, then resolved and quantified using an Applied Biosystems DNA sequencer (ABI 3730xl) equipped with a 36-cm capillary array and nanoPOP-7 polymer.

### Stopped-flow kinetic assay

Stopped-flow experiments were performed as previously described with minor modifications. In this experiment, position 16 in the NTS of the spacer portion of the DNA was labeled with the fluorophore 1,3-diaza-2-oxophenoxazine (tC°) such that, upon R-loop completion, there was an increase in fluorescence as the fluorophore went from a quenched state in the double-stranded form in target DNA to a fluorescent state when it became single stranded as the R-loop is formed. Then, 100–150 nM active-site Cas12f–gRNA complex was mixed with 100 nM tC°-labeled DNA substrate at 46 or 37 °C using an AutoSF-120 stopped-flow instrument (KinTek Corporation, Austin, TX). Excitation was at 367 nm and emission was monitored with a 445-nm filter with a 20-nm bandpass (Semrock).

### Global fitting of kinetic data

Stopped-flow and cleavage kinetic data were globally fitted to the model shown in Extended Data Fig. 10 with KinTek Explorer software version 11 (KinTek Corporation, Austin, TX). Rates shown in black were locked at the values listed in the fitting, while other parameters were allowed to float. For parameters where only an equilibrium constant and a lower limit on a rate were obtained, the parameter with a lower limit listed in Extended Data Fig. 10 was locked at the lower limit in the fitting and the other rate constant for that step was allowed to float. The output for experiments to measure cleavage was modeled simply as the sum of products for each cleavage. For example, for TS cleavage, the products were defined according to our kinetic model as the sum of all species containing P2: EDP2 + EDP1P2. NTS cleavage products were defined as the sum of all species containing P1: EDP1 + XDP1 + EDP1P2. The fluorescence signal for R-loop formation was modeled as a weighted sum of all species contributing to net fluorescence. Specifically, the fluorescence output was modeled as $a$(D + ED +EDx + $b$(EDR + EDP2 + EDP1 + EDP1P2 + XDP1)). The scaling factors, $a$ and $b$, were derived as variable parameters in globally fitting the data. FitSpace confidence contour analysis was performed to define the lower and upper limits for each kinetic parameter and to establish that all parameters (Supplementary Fig. 8), including scaling factors, were well constrained according to this minimal model.

### Reporting summary

Further information on research design is available in the Nature Portfolio Reporting Summary linked to this article.

### Data availability

Structures of Al3Cas12f state I and state II, OsCas12f state I, state II and state III and RhCas12f were deposited to the EM Data Bank with accession codes EMD-49954, EMD-49957, EMD-49959, EMD-49956, EMD-49958 and EMD-49955, respectively. Associated atomic coordinates were deposited to the Protein Data Bank with accession codes 9NZO, 9NZR, 9NZT, 9NZQ, 9NZS and 9NZP, respectively. Protein, gRNA sequences and spacer sequences for nucleases reported here are available in Supplementary Data 1. Source data are provided with this paper.

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

## Acknowledgements

We thank J. Bravo for help with cryo-EM data collection and processing, D.W.T. lab members, K. Senger (Metagenomi Therapeutics) and P. Szymanski (Metagenomi Therapeutics) for valuable discussion and A. Brilot and E. Schwartz (Sauer Structural Biology Laboratory, University of Texas at Austin) for cryo-EM assistance. We thank M. MacKillop and C. Ethridge for mRNA synthesis of the nucleases. We also thank E. Thompson for her thoughtful comments on the paper. This work was supported, in part, by a sponsored research agreeement with Metagenomi Thereaputics (to D.W.T.) and a National Institutes of Health grant R35GM138348 (to D.W.T.). The content is solely the responsibility of the authors and does not necessarily represent the

official views of the National Institutes of Health. Computational resources for this work were supported by the Welch Foundation grant F-1938 (to D.W.T.).

## Author contributions

K.G. and R.F.O. performed the cryo-EM data preparation, collection, structural determination and analysis, as well as the biochemical, kinetic and in vivo assays. P.B.M.C. and C.J.C. performed the metagenomic discovery and phylogenetic characterization. P.B.M.C. performed the phylogenetic and structural analysis of the REC domain. L.G.-O., L.M.A. and D.T.C. designed and performed the in vitro assays. L.G.-O. conducted the assays to test spacer length and contributed to the design of protein variants for engineering. M.B. and R.C.L. performed and analyzed the in vivo experiments. D.T.C. and M.S.W. purified the proteins. L.M.A. analyzed the bioinformatic data. N.C.T. contributed to the structural analyses and led the structure-guided engineering. M.M.H., M.S.W., N.M.A. and I.K. assisted with the biochemistry, kinetics and in vivo assays. T.L.D. and K.A.J. performed and interpreted the kinetic analysis and global fitting. K.G., R.F.O., P.B.M.C. and T.L.D. wrote the paper. D.W.T., D.S.A.G., C.N.B., L.M.A. and C.T.B. supervised the project. D.W.T. and C.T.B. provided the resources and funding for the work. All authors reviewed, edited and approved the paper.

## Competing interests

P.B.M.C., C.J.C., L.G.-O., L.M.A., D.T.C., M.B., R.C.L., N.C.T., D.S.A.G., C.N.B. and C.T.B. are or were employees of Metagenomi Therapeutics and inventors on pending patent applications. K.A.J. is the President of KinTek, which provided the AutoSF-120 stopped-flow instrument and the KinTek Explorer software used in this study. D.W.T. has a sponsored research agreement with Metagenomi. The other authors declare no competing interests.

## Additional information

**Extended data** is available for this paper at https://doi.org/10.1038/s41594-026-01788-6.

**Correspondence and requests for materials** should be addressed to David W. Taylor.

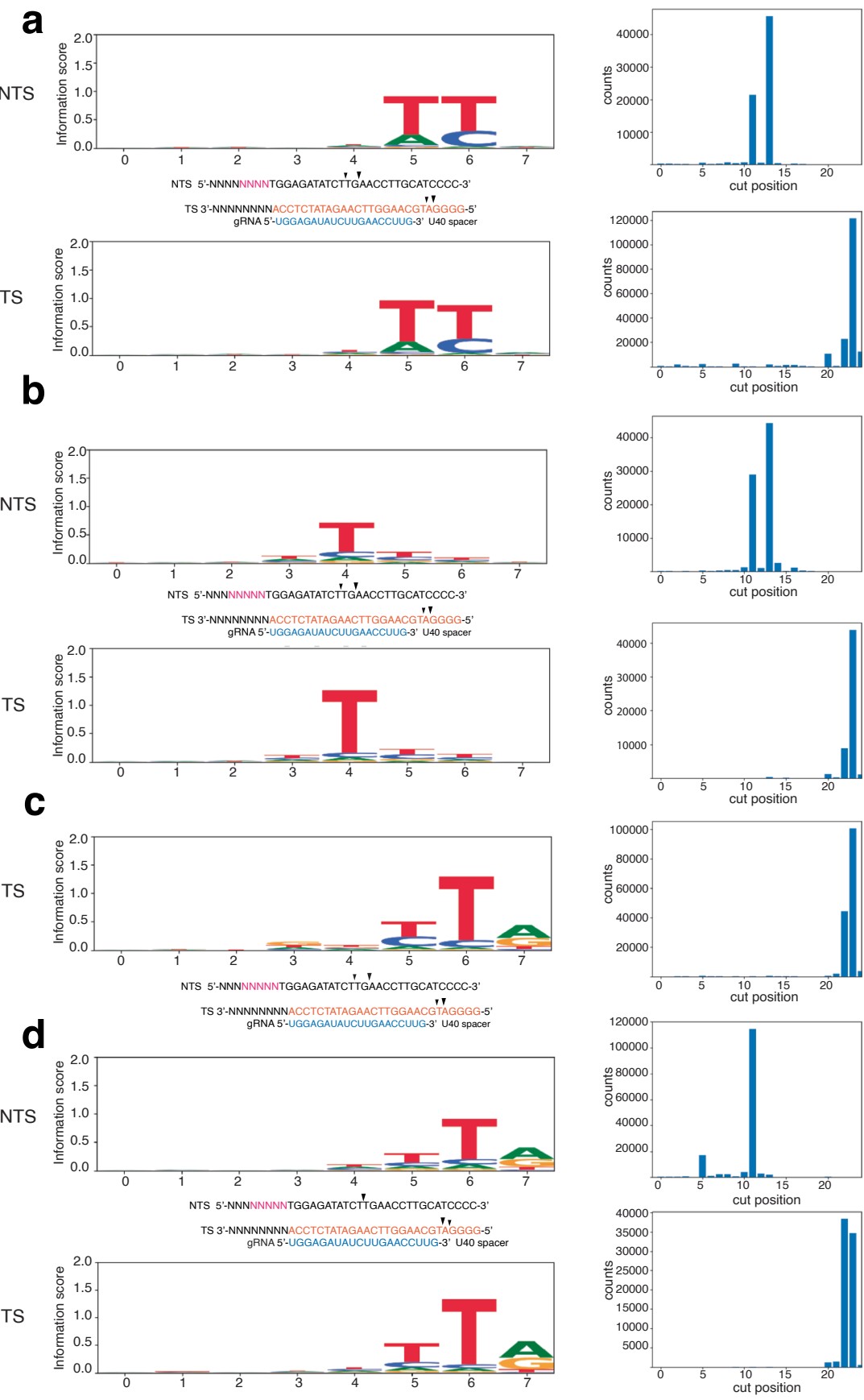

**Extended Data Fig. 1 | See next page for caption.**

**Extended Data Fig. 1 | SeqLogos, cut site schematics with 20 nt U40 spacer and cut site histograms. a**. Cas12f-MG119-1. **b**. Cas12f-MG119-2. **c**. Cas12f-MG119-3. **d**. Al3Cas12f. PAM sequences (pink Ns in schematics) were confirmed by NGS and represented as SeqLogos. SeqLogos were generated from separate cleavage reactions treated with the Klenow fragment or the Mung Bean nuclease to confirm the target strand (TS) cut site or the non-target strand (NTS) cut site, respectively as shown in Supplementary Fig. 2. As expected, the PAM sequence was the same with either treatment. Histograms show the number of NGS reads (counts) mapped to the cleavage site on each strand. Carrots over sequences in the cut sites schematics denote the cut sites. Cas12f-MG119-3 cut site on the NTS was less than a 1000 reads, therefore the SeqLogo and histogram are not shown, however the assumed cut sites are indicated with gray carrots. The cut site on the TS was determined to be 22, 23 and 11, 13 on the NTS based on the number of reads mapped at these locations. In the cut site schematics, the U40 spacer sequence is shown in blue font, and the target DNA sequence is shown in orange font.

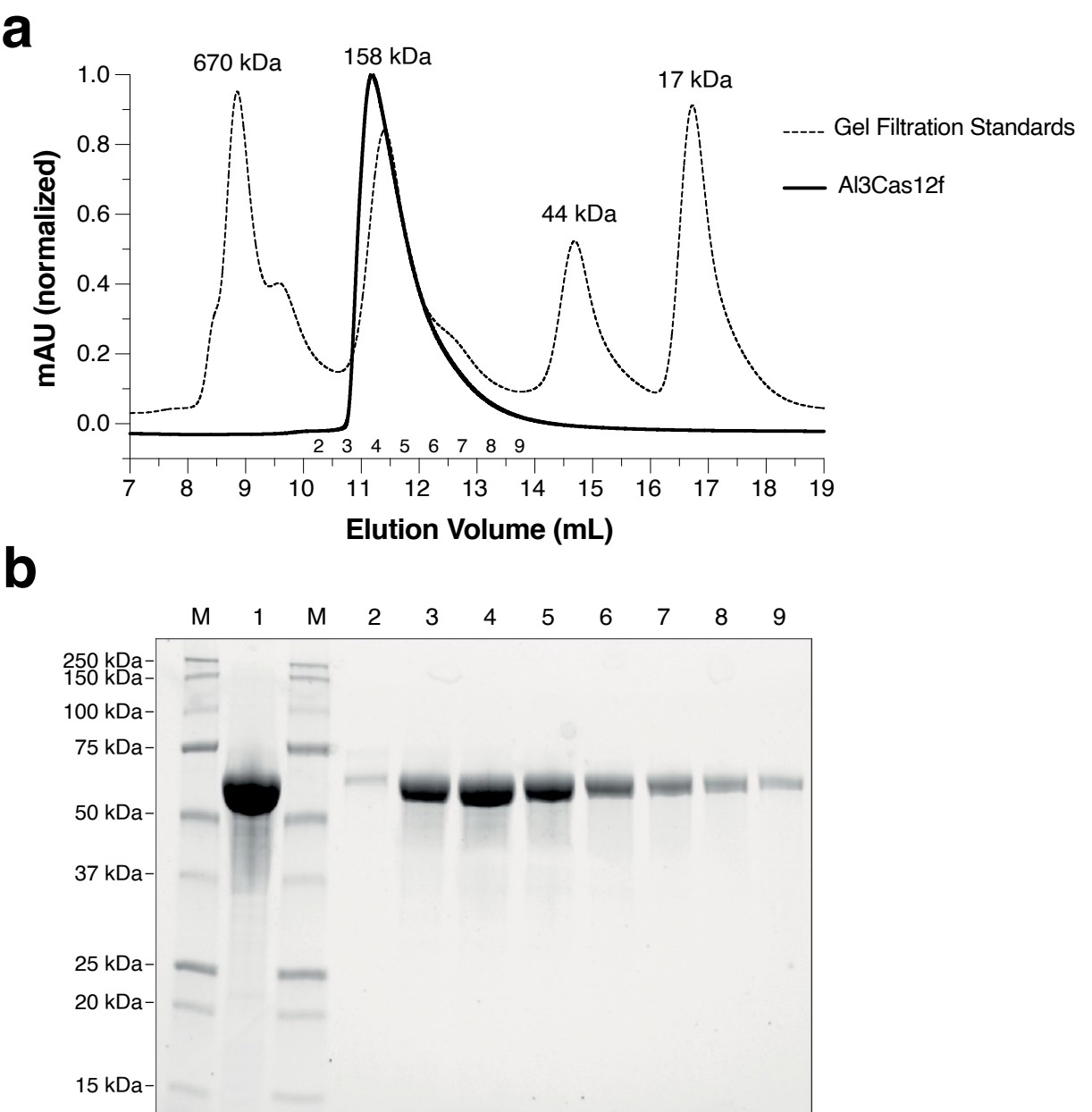

**Extended Data Fig. 2 | Al3Cas12f protein purification. a.** S200i 10 300 SEC elution trace of Al3Cas12f (dark line) overlayed with elution trace of BioRad Gel Filtration Standards (dashed line, Bio-Rad Catalog #1511901). The Al3Cas12f peak runs elutes around 11.5 mL post-injection, similar to the second peak from the gel filtration standards (bovine γ-globulin, approx. 158,000 Da). **b.** SDS-PAGE gel of Al3Cas12f protein purification steps. Sample before (1) and after SEC (2-9). Samples 2–9 correspond to consecutive 0.5 mL fractions collected from elution volume 10 mL to 13.5 mL. M, Marker (Bio-Rad Catalog #1610363).

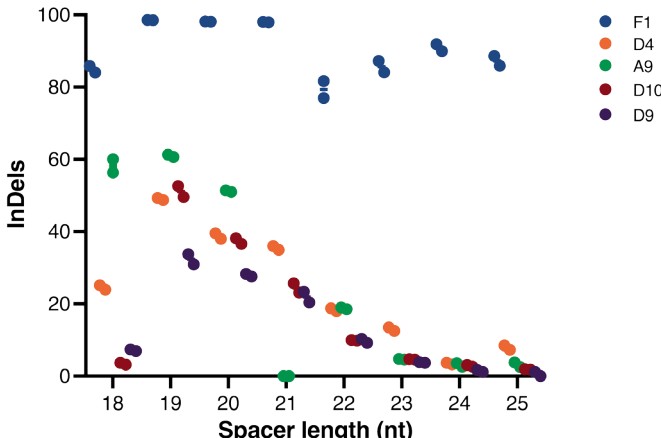

**Extended Data Fig. 3 | Editing efficiency with different spacer lengths and the 134 nt Al3Cas12f gRNA scaffold.** Editing efficiency in K562 cells is shown for guides targeting AAVS1 (F1 and D4) and SOD1 (A9, D9, D10) that had previously displayed a range of editing efficiencies with a spacer length of 20 nt and a truncated version of the gRNA scaffold. Points represent the individual data (n = 2 independent biological replicates). Lines represent the mean value. Overall, guides with the 19 nt spacer displayed the highest levels of editing efficiency in this assay. The guide with the highest editing efficiency across all spacer lengths was AAVS1-F1. See Methods for further details. Source data are provided in Source Data file.

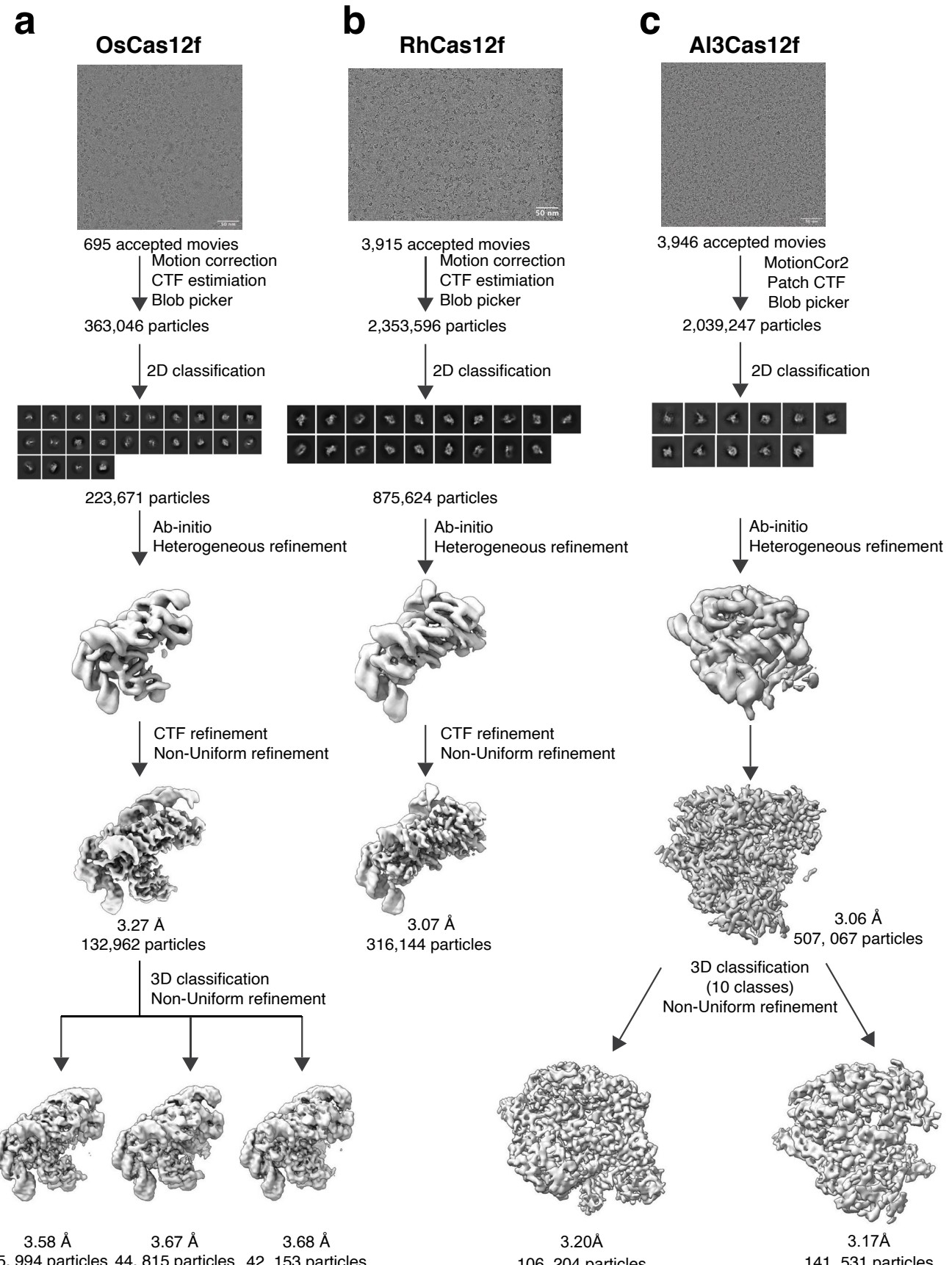

**Extended Data Fig. 4 | Cryo-EM data processing workflow for Cas12f ternary complexes.** Data processing workflow for **a**. OsCas12f. **b**. RhCas12f. **c**. Al3Cas12f.

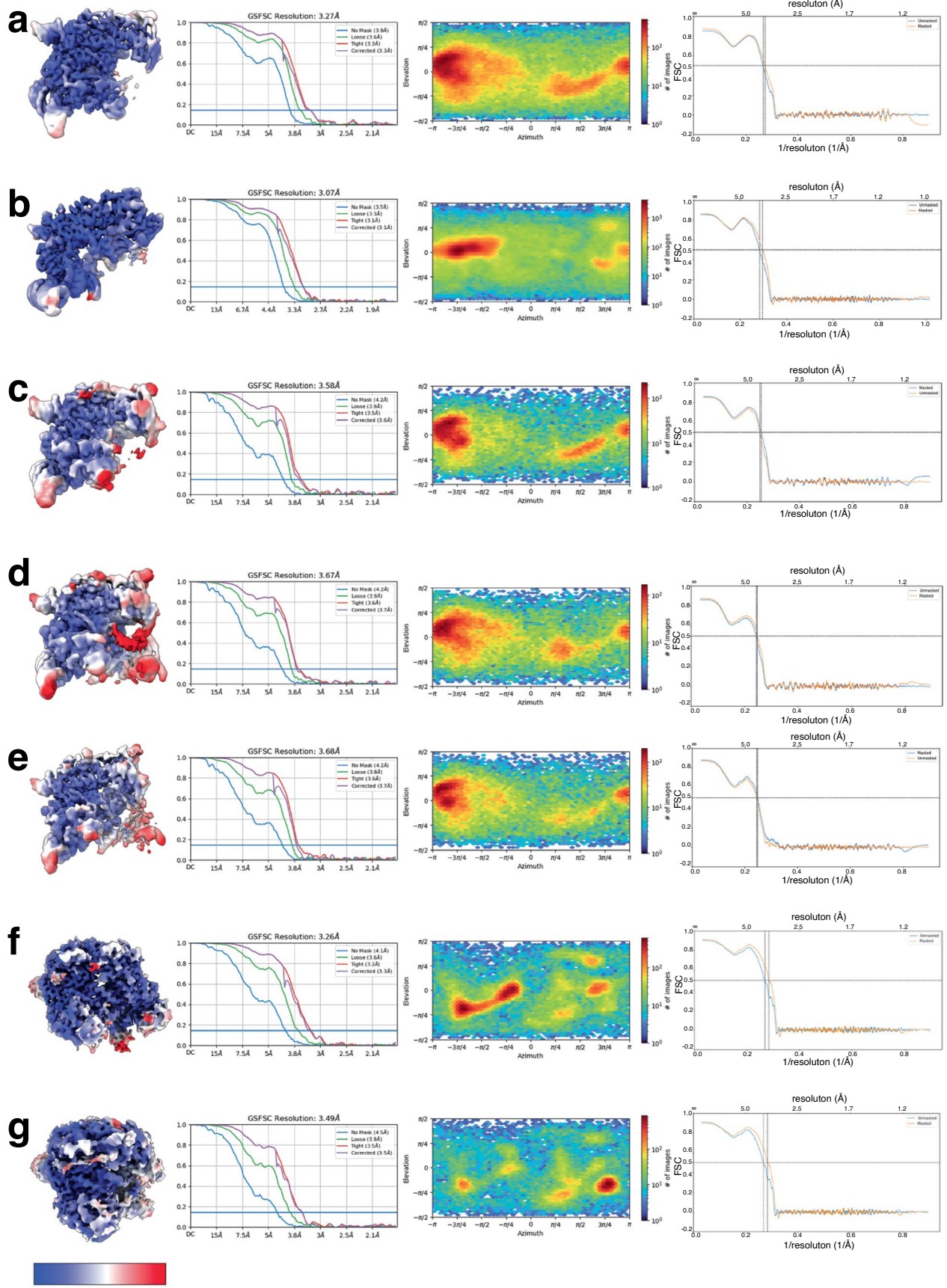

**Extended Data Fig. 5 | Cryo-EM data analysis of Cas12f complexes. a-g.** Cryo-EM data analysis for OsCas12f (**a**), RhCas12f (**b**), OsCas12f intermediate state I (**c**), state II (**d**), state III (**e**), Al3Cas12f intermediate state I (**f**) and state II (**g**). Left to right panels show local resolution maps; gold-standard Fourier Shell Correlation (FSC) curves with resolution reported at FSC = 0.143; Euler angle distribution plots of particle orientations; map-to-model FSC curves.

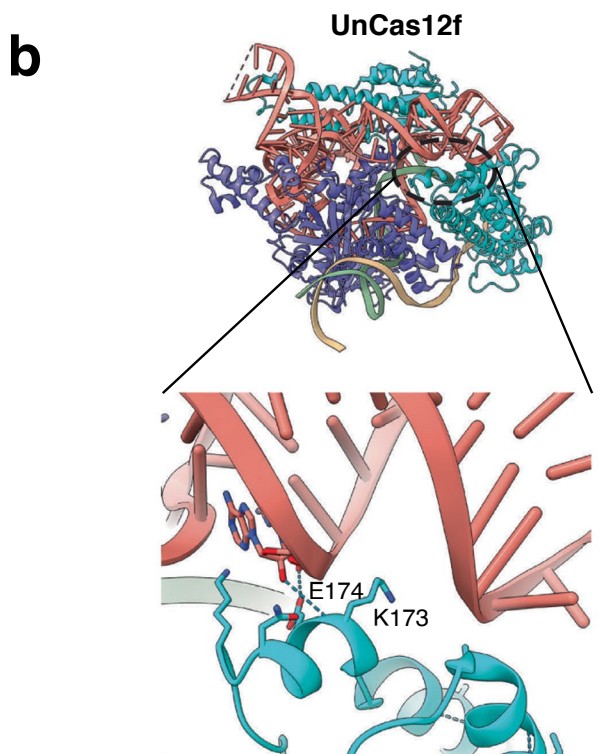

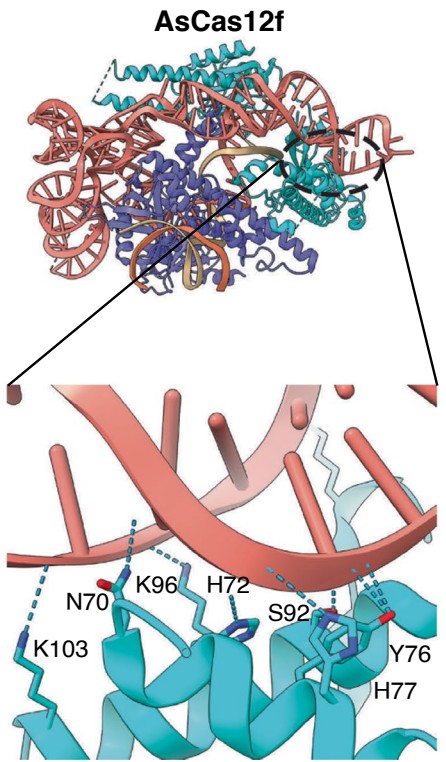

**Extended Data Fig. 6 | Comparative analysis of gRNA architectures across Cas12f orthologs. a**. gRNA scaffolds of AsCas12f (PDB: 8J12), OsCas12f, RhCas12f, Al3Cas12f, UnCas12f (PDB: 7L49), and engineered AsCas12f (PDB: 8J1J). Stems, spacers, and unresolved regions (dashed lines) are labeled. **b**. Structures of UnCas12f and AsCas12f ternary complexes. Insets highlight hydrogen-bond interactions (blue dashed lines) between the gRNA Stem 2 terminus and Molecule 2.

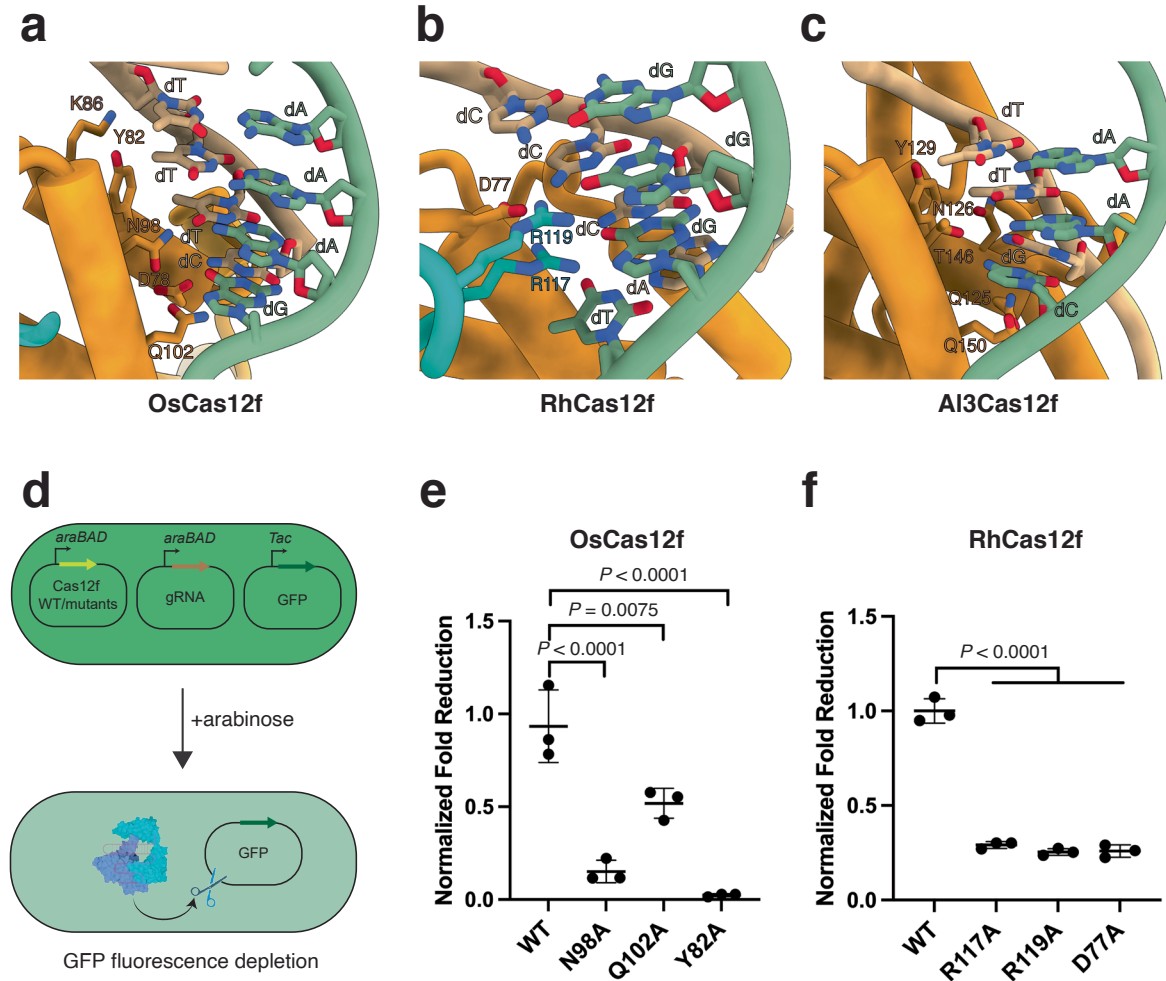

**Extended Data Fig. 7 | PAM recognition by Os-, RhCas12f and Al3Cas12f.**
Interactions between the PAM duplex and OsCas12f (**a**), RhCas12f (**b**), and
Al3Cas12f (**c**). **d**. Schematic of the GFP depletion assay. **e-f**. In vivo activity of
wild-type (WT) and PAM-recognition mutants of OsCas12f (**e**) and RhCas12f (**f**).
Data represent mean ± SD (n = 3 independent biological replicates). Significance
determined by one-way ANOVA. Source data are provided in Source Data file.

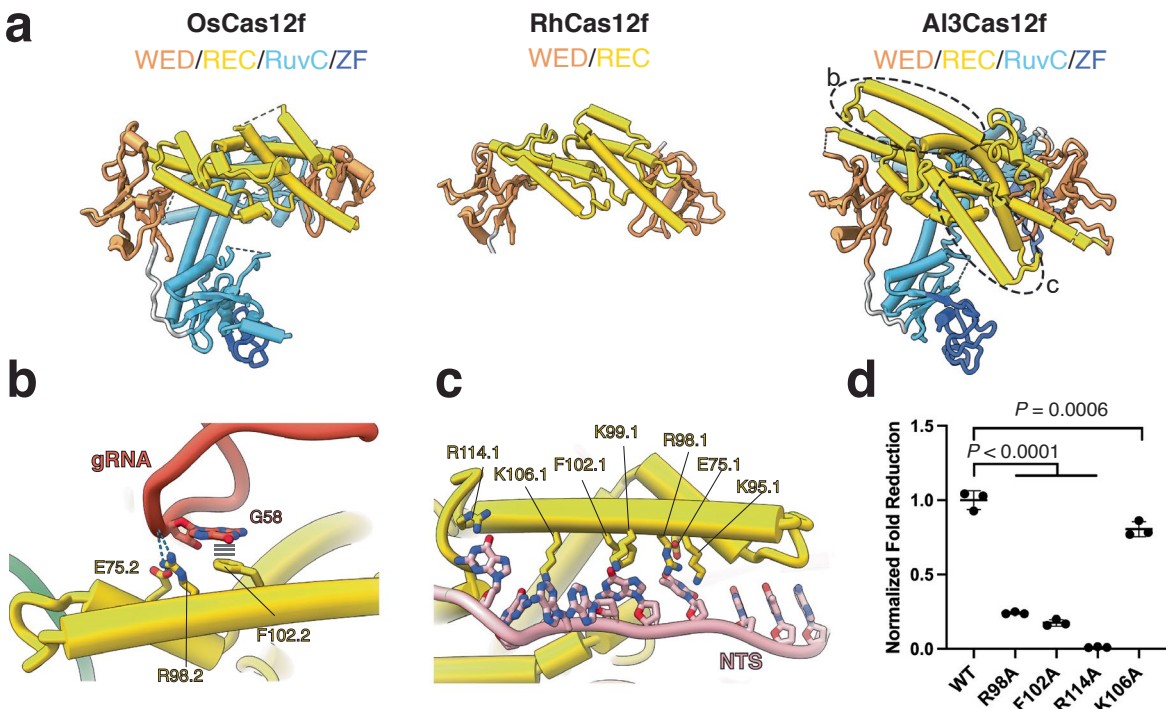

**Extended Data Fig. 8 | Extended helices in Al3Cas12f REC domain stabilize gRNA and non-target strand DNA. a**. Protein structure comparison of OsCas12f, RhCas12f, and Al3Cas12f. **b**. Interaction between the extended helix in Al3Cas12f Molecule 2 and the gRNA Stem 2 terminus. **c**. Interaction between the extended helix in Al3Cas12f Molecule 1 and the NTS DNA. **d**. In vivo activity of WT and helix mutants of Al3Cas12f in GFP depletion assay. Data represent mean ± SD (n = 3 independent biological replicates). Significance determined by one-way ANOVA. Source data are provided in Source Data file.

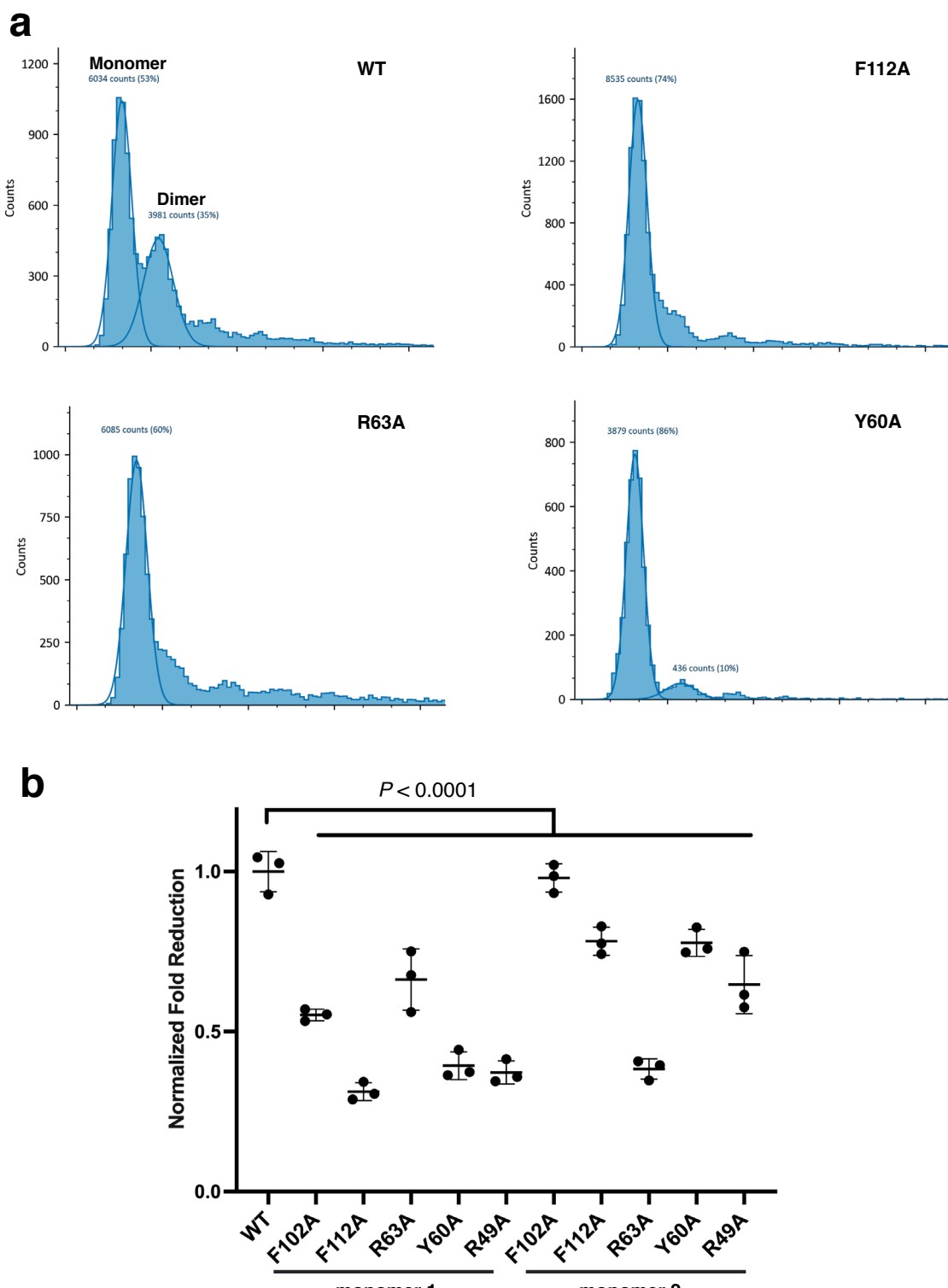

**Extended Data Fig. 9 | Stability assessment of Al3Cas12f dimer interface.**
**a**. Oligomeric state analysis (dimer:monomer ratio) of Al3Cas12f using mass photometry. The WT variant displays a higher dimer: monomer ratio than the single amino acid mutants. **b**. GFP depletion assays for single amino acid mutations of Al3Cas12f covalent dimer. Data represent mean ± SD (n = 3 independent biological replicates). Significance determined by one-way ANOVA. Source data are provided in Source Data file.

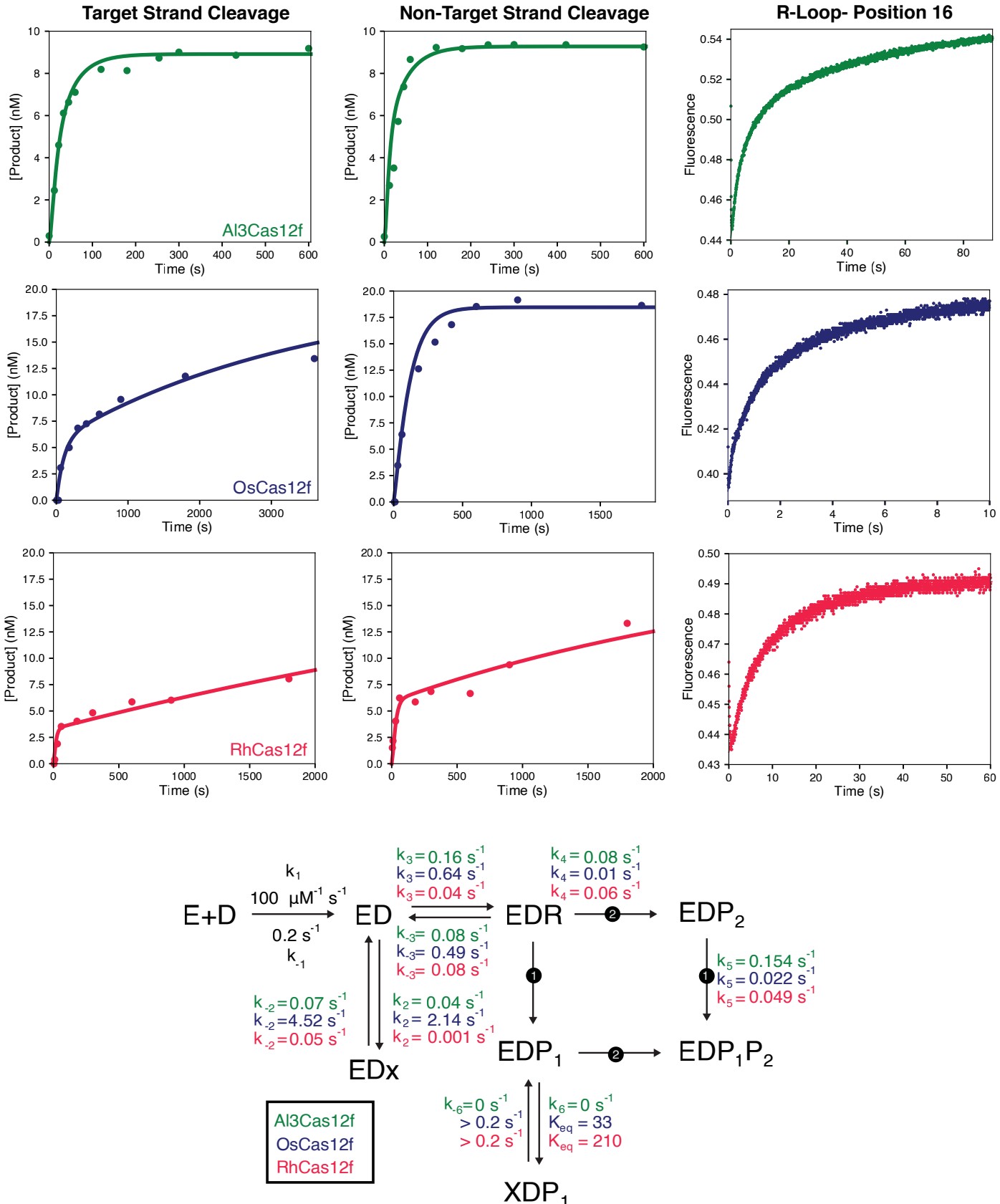

**Extended Data Fig. 10 | Kinetics of cleavage and R-loop formation by Cas12f variants.** Top: Cleavage and R-loop formation kinetics for Al3Cas12f, OsCas12f, and RhCas12f are shown in green, blue, and red, respectively. Bottom: Kinetic scheme generated in KinTek for DNA binding, R-loop formation, and cleavage for each Cas12f variant. Rate constants in black were locked at the values shown in the fitting. Source data are provided in Source Data file.

# Reporting Summary

## Statistics

For all statistical analyses, confirm that the following items are present in the figure legend, table legend, main text, or Methods section.

| n/a | Confirmed | |
|---|---|---|
| ☐ | ☒ | The exact sample size (*n*) for each experimental group/condition, given as a discrete number and unit of measurement |
| ☒ | ☐ | A statement on whether measurements were taken from distinct samples or whether the same sample was measured repeatedly |
| ☐ | ☒ | The statistical test(s) used AND whether they are one- or two-sided *Only common tests should be described solely by name; describe more complex techniques in the Methods section.* |
| ☒ | ☐ | A description of all covariates tested |
| ☒ | ☐ | A description of any assumptions or corrections, such as tests of normality and adjustment for multiple comparisons |
| ☐ | ☒ | A full description of the statistical parameters including central tendency (e.g. means) or other basic estimates (e.g. regression coefficient) AND variation (e.g. standard deviation) or associated estimates of uncertainty (e.g. confidence intervals) |
| ☐ | ☒ | For null hypothesis testing, the test statistic (e.g. *F*, *t*, *r*) with confidence intervals, effect sizes, degrees of freedom and *P* value noted *Give P values as exact values whenever suitable.* |
| ☒ | ☐ | For Bayesian analysis, information on the choice of priors and Markov chain Monte Carlo settings |
| ☒ | ☐ | For hierarchical and complex designs, identification of the appropriate level for tests and full reporting of outcomes |
| ☒ | ☐ | Estimates of effect sizes (e.g. Cohen's *d*, Pearson's *r*), indicating how they were calculated |

*Our web collection on statistics for biologists contains articles on many of the points above.*

## Software and code

Policy information about availability of computer code

| Data collection | Grids were screened with SerialEM on a FEI Glacios cryo-TEM. For Cas12f-MG119-28 and OsCas12f ternary complexes, images were collected on a FEI Glacios cryo-TEM equipped with a Falcon 4 detector with a pixel size of 0.933 Å, while the images of the RhCas12f ternary complex were collected on a FEI Titan Krios cryo-TEM equipped with a Gatan K3 direct electron detector with a pixel size of 0.8332 Å. The defocus range was set to -1.5 to -2.5 μm. Motion correction, contrast transfer function (CTF) estimation and particle picking were carried out in cryoSPARC live v4.0. |
|---|---|
| Data analysis | All subsequent data processing was carried out in cryoSPARC v4.4. The Cas12f protein structure predicted by AlphaFold2 was fitted into the ternary complex map as a rigid body in ChimeraX. gRNA was modeled based on secondary structure prediction and gRNA architecture of AsCas12f. The model was manually refined in COOT 43, and automatically refined by real_space_refine in PHENIX. Cas12f-MG119-28 was built using Cosmic2 ModelAngelo. The model was subsequently manually refined using COOT and Isolde, and automatically refined by real_space_refine in PHENIX. All structural figures were generated using ChimeraX v1.7.1. |

For manuscripts utilizing custom algorithms or software that are central to the research but not yet described in published literature, software must be made available to editors and reviewers. We strongly encourage code deposition in a community repository (e.g. GitHub). See the Nature Portfolio guidelines for submitting code & software for further information.

## Data

Policy information about [availability of data](availability of data)

All manuscripts must include a [data availability statement](data availability statement). This statement should provide the following information, where applicable:

- Accession codes, unique identifiers, or web links for publicly available datasets
- A description of any restrictions on data availability
- For clinical datasets or third party data, please ensure that the statement adheres to our [policy](policy)

Structures of the Cas12f-MG119-28 State I, State II, OsCas12f State I, State II, State III, RhCas12f have been deposited in the EMDB with accession codes: EMD-49954, EMD-49957, EMD-49959, EMD-49956, EMD-49958, EMD-49955, respectively. Associated atomic coordinates were deposited to PDB with accession codes: 9NZO, 9NZR, 9NZT, 9NZQ, 9NZS, 9NZP, respectively. Protein and guide RNA sequences for nucleases reported here are available in Supplemental Data tables.

## Research involving human participants, their data, or biological material

Policy information about studies with [human participants or human data](human participants or human data). See also policy information about [sex, gender (identity/presentation), and sexual orientation](sex, gender (identity/presentation), and sexual orientation) and [race, ethnicity and racism](race, ethnicity and racism).

| | |
|---|---|
| Reporting on sex and gender | Not applicable. |
| Reporting on race, ethnicity, or other socially relevant groupings | Not applicable. |
| Population characteristics | Not applicable. |
| Recruitment | Not applicable. |
| Ethics oversight | Not applicable. |

Note that full information on the approval of the study protocol must also be provided in the manuscript.

# Field-specific reporting

Please select the one below that is the best fit for your research. If you are not sure, read the appropriate sections before making your selection.

☒ Life sciences   ☐ Behavioural & social sciences   ☐ Ecological, evolutionary & environmental sciences

For a reference copy of the document with all sections, see [nature.com/documents/nr-reporting-summary-flat.pdf](nature.com/documents/nr-reporting-summary-flat.pdf)

# Life sciences study design

All studies must disclose on these points even when the disclosure is negative.

| | |
|---|---|
| Sample size | A sample size of three independent biological replicates were required to derive statistical measurements. A sample size of two was chosen for initial screening purposes to show a general trend. |
| Data exclusions | No data was excluded. |
| Replication | All experiments were replicated at least three times with similar results. |
| Randomization | Particle orientations are randomized during processing. |
| Blinding | No blinding was performed. The experiments were observational and not subject to bias. |

# Reporting for specific materials, systems and methods

We require information from authors about some types of materials, experimental systems and methods used in many studies. Here, indicate whether each material, system or method listed is relevant to your study. If you are not sure if a list item applies to your research, read the appropriate section before selecting a response.

## Materials & experimental systems

| n/a | Involved in the study |
|-----|-----------------------|
| ☒ | ☐ Antibodies |
| ☐ | ☒ Eukaryotic cell lines |
| ☒ | ☐ Palaeontology and archaeology |
| ☒ | ☐ Animals and other organisms |
| ☒ | ☐ Clinical data |
| ☒ | ☐ Dual use research of concern |
| ☒ | ☐ Plants |

## Methods

| n/a | Involved in the study |
|-----|-----------------------|
| ☒ | ☐ ChIP-seq |
| ☒ | ☐ Flow cytometry |
| ☒ | ☐ MRI-based neuroimaging |

## Eukaryotic cell lines

Policy information about cell lines and Sex and Gender in Research

| | |
|---|---|
| Cell line source(s) | K562 cells were used in cell editing experiments. |
| Authentication | Cells were purchased from ATTC and cultured using manufacturer protocols. |
| Mycoplasma contamination | Cell lines were not tested for Mycoplasma contamination. |
| Commonly misidentified lines (See ICLAC register) | Non used. |

## Plants

| | |
|---|---|
| Seed stocks | Not applicable. |
| Novel plant genotypes | Not applicable. |
| Authentication | Not applicable. |

