## [Peer Review File · Nature Structural & Molecular Biology]

Comparative characterization of Cas12f orthologs reveals mechanistic features underlying enhanced genome editing efficiency

Corresponding Author: Dr David Taylor

Version 0:

Decision Letter:

26th Jun 2025

Dear Dr. Taylor,

Thank you again for submitting your manuscript "Comparative characterization of Cas12f orthologs reveals mechanistic features underlying enhanced genome editing efficiency". I apologize for the delay in sharing our decision. I am writing to let you know that we have decided to send your manuscript for peer review, but some points require your attention before we can proceed with peer review.

I am re-opening the manuscript submission system for you to resubmit your manuscript with all associated files needed for peer review directly, within 2-3 business days if possible. Please follow the link at the bottom of this email to upload the documents listed below. If you have any issues, please reach out to us before completing the submission.

1- Thanks for providing official wwPDB validation reports for newly described atomic structures, as per journal policy. We also request that authors provide cryo-EM maps, half-maps and models to help the reviewers in assessing the work. We recommend the use of figshare in our system, which allows for provision of anonymous access links for the referees (<https://www.springernature.com/gp/authors/research-data/figshare-integration>). Alternatively, please upload .zip folders directly with the submission. To ensure the ease of reviewer access to the data, please specify in the Data Availability section where the files can be found (i.e., provide a figshare link or direct the reader to the manuscript files).

2- We want to ensure that the methods and statistics reporting in our papers are of the highest quality. To that end, we ask authors to fill out a Reporting Summary that collects information on experimental design and reagents. Thanks for already providing them but we ask you to please double check their validity and that the manuscript matches the information in them.

3- Lastly, I would like to kindly request that you provide the code used to analyse the data to the reviewers, if newly developed (unpublished) code was used in the work. For the reviewers to evaluate the work adequately, they must be able to test the software/review the code themselves. If you have not yet provided the software, we therefore request that you provide a single compressed zip file containing the software with a readme.txt file or other user manual containing complete instructions for installing and running the software. If appropriate, please provide example data and expected output. Sufficient material should be provided for referees to directly test the performance of the software/algorithm. If the software and materials are small enough to fit in a single compressed zip file under 6MB in size, you may email this file directly to me. If the zip file is between 6 MB and 200 MB, you may upload it to our file transfer site. If necessary, a second zip file up to 200 MB in size can be used to supply the example data. Please let me know if you need to use this option and I'll send you further details. Alternatively, you can upload the code to GitHub and provide us with the link.

Please fill out and return to me the code and software submission checklist that will be made available to editors and reviewers during manuscript assessment. Please note that this form is a dynamic 'smart pdf' and must therefore be downloaded and completed in Adobe Reader, instead of opening it in a web browser.
<https://www.nature.com/documents/nr-software-policy.pdf>

Please use the link below to submit the files. **Please also remember to move forward all other files associated with this version of the paper.**

Link Redacted

We look forward to seeing the revised manuscript and thank you for the opportunity to review your work. Once we have this information we will start trying to recruit reviewers.

Sincerely,

Dimitris Typas
Senior Editor
Nature Structural & Molecular Biology
ORCID: 0000-0002-8737-1319

Version 1:

Decision Letter:

1st Aug 2025

Dear Dr. Taylor,

Thank you again for submitting your manuscript "Comparative characterization of Cas12f orthologs reveals mechanistic features underlying enhanced genome editing efficiency". I apologise for the delay in responding, which resulted from the difficulty in obtaining and timely discussing referee reports due to travelling. Nevertheless, we now have comments (below) from the 3 reviewers who evaluated your paper. In light of these reports, we remain interested in your study and would like to see your response to the comments of the referees, in the form of a revised manuscript.

You will see that the experts appreciate the potential of the newly identified Cas12f-MG119-28, but they raise numerous concerns that must be comprehensively addressed in a revised manuscript. In particular, multiple reviewers note that further benchmarking in mammalian cells is necessary and Reviewer #3 correctly mentions that the performance in mammalian cells needs to be assessed also in comparison to existing editors and with respect to off-target effects and indels. Mechanistically, the referees request further support for the claim that dimerisation controls editing efficiency and provide pertinent experimental guidelines on how to do so. Functionally, the reviewers request further investigation of the effects of guide length and sequence on editing efficiency and Reviewer #1 brings up several very important points about necessary figure improvements and clarifications.

Please be sure to address/respond to all concerns of the referees in full in a point-by-point response and highlight all changes in the revised manuscript text file. If you have comments that are intended for editors only, please include those in a separate cover letter.

We expect to see your revised manuscript within 4 months though this deadline is flexible. If you cannot send it within this time, please contact us for an extension which we will likely grant, provided that no similar work has been accepted for publication at NSMB or published elsewhere.

Reporting Summary:
<https://www.nature.com/documents/nr-reporting-summary.pdf>

-- that unprocessed scans are clearly labelled and match the gels and western blots presented in figures. Please note that all key data shown in the main figures as cropped gels or blots should be presented in uncropped form, with molecular weight

markers. While these data can be displayed in a relatively informal style, they must refer back to the relevant figures. These data should be submitted as source data with the last revision, prior to acceptance.

-- that control panels for gels and western blots are appropriately described as loading on sample processing controls

-- all images in the paper are checked for duplication of panels and for splicing of gel lanes.

-- For any revision that includes light microscopy data, we ask our authors to please include a completed light microscopy reporting table [https://www.nature.com/documents/Light_microscopy_reporting_table.xlsx] to ensure the methods are described thoroughly. The table will be available to reviewers and ultimately published should the manuscript be accepted at the journal.

EXTENDED DATA FIGURES

Data availability: this journal strongly supports public availability of data. All data used in accepted papers should be available via a public data repository, or alternatively, as Supplementary Information. If data can only be shared on request, please explain why in your Data Availability Statement, and also in the correspondence with your editor. Please note that for some data types, deposition in a public repository is mandatory - more information on our data deposition policies and available repositories can be found below:

<https://www.nature.com/nature-research/editorial-policies/reporting-standards#availability-of-data>

Link Redacted

Sincerely,

Reviewers' Comments:

Reviewer #1 (Remarks to the Author):

CRISPR-Cas12f nucleases are smaller than other CRISPR enzymes and have been used as miniature genome editors. In this study, Guan et al. identified Cas12f-MG119-28, which shows robust genome editing activity in human cells. Furthermore, the authors performed structural and functional analyses of Cas12f-MG119-28, *Oscillibacter* sp. Cas12f (OsCas12f) and *Ruminiclostridium herbifermentans* Cas12f (RhCas12f). A comparative structural analysis reveals both similarities and differences among these orthologs, providing insights into key mechanistic determinants of Cas12f activity. Overall, this manuscript advances our understanding of the diverse miniature Cas12f nucleases and offers a framework for developing genome-engineering tools with therapeutic potential. However, the quality of some figures is extremely poor and cannot be properly interpreted. Therefore, I am unable to support the publication of the current version of the manuscript. The authors should address the following points for substantial revision.

Major points:

L84: "These results, combined with the observation that two target sites achieved >90% editing at AAVS1 (Fig. 1d), suggest that this nuclease is highly active in its native form."

- This is interesting and surprising. However, the authors should more carefully evaluate the editing activity of Cas12f-MG119-28 by comparing it with other highly active nucleases. At minimum, they should compare its editing efficiency with those of SpCas9 and enAsCas12 (10.1016/j.cell.2023.08.031) across multiple sites in human cells.

L314: "While we show the gRNA of Cas12f-MG119-28 has already been naturally optimized..."

- It would be beneficial to further improve the editing activity of Cas12f-MG119-28 through structure-guided engineering.

Points:

L71: "Upon inspection of the genomic regions encoding the nuclease genes, we identified the corresponding tracrRNAs and crRNAs. Cas12f-MG119 representatives require large sgRNAs (129 - 156 nt) as empirically confirmed (Fig. 1b, Suppl. Fig. 1 and Suppl. Data 2)."

- Fig. 1b and Suppl. Fig. 1 lack sufficient information and should be substantially revised. In addition, the sequences of the crRNA and tracrRNA should be provided.

L74: "We determined that these systems recognize T-rich PAMs using an in vitro cleavage assay of an 8N PAM library (Fig. 1c), and cut the target strand at 20, 22, and 23 nt from the 5' PAM sequence and the non-target strand at 11 and 13 nt (Suppl. Fig. 2 and 3)."

- Since CRISPR nucleases use guide (spacer) segments of various lengths (e.g., 20 nt for SpCas9), the authors should specify the guide length of Cas12f-MG119-28 and examine how guide length affects DNA cleavage efficiency in vitro.

L102: "This enzyme most closely resembles the structure of *Clostridium novyi* (CnCas12f)"

- Including a figure comparing the structures of CnCas12f and Cas12f-MG119-28 would be helpful.

L109: "Since there are limited Cas12f structures available, we aimed to perform a more thorough comparison of several orthologs known to have high levels of genome editing activity, obtaining structures of OsCas12f and RhCas12f bound to their respective guides and target DNA"

- The rationale for selecting OsCas12f and RhCas12f for structural and functional comparison should be explained. Also, the sequence identities between Cas12f-MG119-28 and these orthologs should be provided.

L124: "their RNA is much shorter than their counterpart since they contain only four stem loops"

- Stem 1 is not classified as a stem loop; clarify this point.

L194: "Cas12f-MG119-28 displayed two different intermediates, both with a fully formed 20 bp R-loop: a pre-product state prior to RuvC.2 docking (State I) and a post-product state with a fully docked RuvC.2 domain at the PAM-distal end of the spacer (State II)"

- Do these states correspond to pre- and post-cleavage states? Clarify this.

L213: "We show that this lid is essential for cleavage by mutating E334-R343 to alanine, preventing the lid from undergoing the rearrangements necessary for cleavage"

- Why does the alanine mutation in E334-R343 prevent the rearrangement of the lid region?

L223: "In contrast to OsCas12f, both structures of Cas12f-MG119-28 showed the lid in an open conformation, indicating that Cas12f-MG119-28 can adapt its cleavage-active conformation prior to RuvC.2 docking."

- It would be helpful to include an explanation for why, unlike OsCas12f, Cas12f-MG119-28 can adopt an active conformation before RuvC.2 docking.

L265: "For RhCas12f, approximately 45% partitioned towards non-target strand cleavage, essentially showing little preference towards which strand is cleaved first."

- How can RhCas12f cleave the target strand prior to the non-target strand? Please provide structural explanations.

Fig. 1b: This figure is of very poor quality and lacks clarity. Please show the sequences and indicate the spacer and repeat regions of the crRNA and the tracrRNA region.

Fig. 1c: Unnecessary labels (such as 119-28-U40... etc.) should be removed for clarity.

Fig. 1d: Did the authors measure editing efficiency only once? Replicates (triplicates or more) are needed for statistical validity.

Fig. 1e: Compare the activity of MG119-28 with that of enAsCas12f (10.1016/j.cell.2023.08.031) in E. coli. Also, ensure consistent notation: "As. Cas12f" etc. should be correctly written as "AsCas12f." etc.

Fig. 2a: It would be better to include domain schematics of Cas12f-MG119-28, OsCas12f, and RhCas12f, with their residue numbering, highlighting similarities and differences.

Fig. 2b-d: Show the protein names for clarity.

Fig. 3a: Indicate the crRNA and tracrRNA regions, as well as the tetraloop linking them.

Fig. 3b: Add labels for the protein domains.

Fig. 3c: Depict hydrogen bonds using dashed lines for clarity.

Fig. 4f: Clarify how each panel relates to those in e. Currently, it is confusing that b, d, and f contain multiple panels. Depict hydrogen bonds with dashed lines.

Supplementary Figure 1: The figure quality is very poor and not interpretable. Please indicate what each band represents (DNA substrate, cleavage product, sgRNA). Include no-enzyme control. Add schematics of sgRNA designs. Briefly explain the reaction conditions (incubation time, PAM sequences, guide lengths, visualization methods). Clarify what labels like "U50_A2" mean and explain the unlabeled lanes. Are the procedures described in the method section?

Supplementary Figure 2: This figure is also of poor quality; please clarify what each band represents. Remove unnecessary lanes and present upper and lower parts separately. Briefly explain reaction conditions in the legend.

Supplementary Figure 3: Explain what the two PAM logos in panels a and b indicate. Why is only one panel shown in c? Include schematics of guide (spacer) sequences, target DNA, and PAMs, along with cleavage sites. Clarify the guide lengths used in the assay.

Supplementary Figure 3 and 4: Merge these figures. The SEC elution profile should be presented before the SDS-PAGE gel. In the SEC data, please label the molecular masses of standards and indicate fractions analyzed by SDS-PAGE.

Supplementary Figure 7: Enlarge labels for better readability

Minor points:

L50: "Although these proteins have yet to reach saturating levels of editing in mammalian cells, rational engineering of Cas12f proteins, as well as their guide RNA (gRNA) scaffolds, have significantly enhanced their gene-editing activity 21,23–25."

- Please cite a recent publication on SpCas12f (<https://doi.org/10.1093/nar/gkaf588>).

L57: "we solved the structures of OsCas12f and RhCas12f, conducted kinetic analyses of all three orthologs..."

- Briefly describe OsCas12f and RhCas12f here, as the introduction is independent from the abstract.

L268: "In previous studies, we have performed magnesium-initiated cleavage experiments where the enzyme, guide RNA, and DNA are preincubated in the absence of magnesium and then magnesium is added to initiate the reaction."

- Please include relevant references.

L297: "Prior to this study, three structures of Cas12f nucleases had been described."

- Include corresponding references.

L299: "Structures of Cas12f-MG119-28, OsCas12f, and RhCas12f in combination with AsCas12f, UnCas12f, and CnCas12f..."

- Provide appropriate citations.

L478: "Approximately 1E+5 cells were transfected"

- Revise to standard notation, e.g., " 1×10^5 cells."

L559: "For Cas12f-MG119-28, 11,080 movies were collected, and 2,039,236 particles were extracted for downstream analysis."

- There are discrepancies with Supplementary Fig. 6, which indicates 2,039,247 particles. Please verify and correct for consistency.

Materials and methods: There are many typos that should be corrected. For example, 100mM -> 100 mM; E.coli -> E. coli (italicized); MgCl₂ (2 should be in subscript); 18000 rpm should be specified as g-force (e.g., "g" units); "micro" (not u), "prime" and "minus sign" should be used correctly and consistently.

Supplementary Figure 11: Please add labels indicating the domains, RNA regions, and DNA regions.

Supplementary Figure 13: Expand the figure legend to include more detailed explanations.

Reviewer #2 (Remarks to the Author):

The manuscript presents a structural and functional characterization of three Cas12f orthologs, with a particular focus on the newly identified Cas12f-MG119-28 variant. The study combines cryo-EM structural determination with extensive biochemical and kinetic analyses to understand the mechanistic basis for the enhanced editing efficiency of Cas12f-MG119-28. The work provides valuable insights into CRISPR-Cas12f working mechanisms that could inform future engineering of compact genome editors. The manuscript is generally well-written and well-organized, though I have several suggestions for improvement as detailed below.

Major Comments

1. The name "Cas12f-MG119-28" is unnecessarily complex and could hinder broader adoption. Consider a simplified designation to align with conventions for ortholog naming.
2. The editing efficiency of the newly identified Cas12f should be benchmarked against larger nucleases, as the authors claim that it is a highly efficient genome editor.
3. Lines 80-84: The manuscript states that editing efficiencies were determined for multiple loci, but Figure 1d lacks error bars or indication of biological replicates. Given this is a key result supporting Cas12f-MG119-28's high activity, please clarify whether the data represents biological replicates.
4. Lines 88-90: The comparison of editing efficiency in E. coli appears to use a single GFP reporter target. As target sequence preferences significantly influence Cas12f variant's activity, I recommend extending the comparison to include multiple target sequences to ensure observed performance differences are generalizable rather than target-specific.
5. Lines 176-179: While alanine substitutions in the REC domain reduce editing activity, Figure 4g data alone cannot establish whether this results from impaired dimerization or other functional disruptions. To strengthen causality, additional characterization (e.g., SEC of mutants to quantify dimer stability) would be valuable, as done for WT protein (Suppl. Fig. 5).
6. The manuscript presents structural and kinetic evidence for divergent activation mechanisms among Cas12f orthologs. Integrating these findings into a mechanistic model figure would improve conceptual clarity.

Minor comments

7. The protein sequences used in the fig. 1a need to be indicated.
8. Lines 197-198: The reason for assigning N-terminal to OsCas12f.1 and C-terminal to OsCas12f.2 in the covalent dimer should be clarified, as this impacts interpretation of activation mechanisms.
9. The unresolved RuvC domains in RhCas12f.1 resemble observations in TnpB (PDB: 8EX9) and Cas12n (PDB: 9J09). A comparative analysis of these systems' conformational flexibility in the discussion section could provide insights into evolutionary constraints governing nuclease domain stability in compact Cas12 systems.
10. Line 613: "intubated" should be corrected to "incubated".

Reviewer #3 (Remarks to the Author):

The study investigates the structural and functional diversity of Cas12f nucleases, focusing on a newly identified ortholog, Cas12f-MG119-28, discovered through metagenomic analysis. The authors compare this nuclease with two other Cas12f variants (OsCas12f and RhCas12f) using cryo-EM, biochemical assays, and kinetic analyses. Four key findings are presented:

1. High editing efficiency: Cas12f-MG119-28 exhibits robust genome editing in human cells, and outperforms other Cas12f

orthologs in *E. coli*.

2. Structural insights: Cryo-EM structures reveal distinct dimerization interfaces, gRNA architectures, and RuvC domain dynamics among the orthologs.

3. Mechanistic diversity: Cas12f-MG119-28 achieves efficient R-loop formation via a stable dimer interface and optimized gRNA, while OsCas12f and RhCas12f rely on different activation mechanisms.

4. Kinetic analysis: Cas12f-MG119-28 shows faster and more efficient DNA cleavage compared to OsCas12f and RhCas12f, attributed to its stable dimerization and pre-formed R-loop.

Overall, this study represents a significant advancement in the CRISPR field, providing deep mechanistic insights into Cas12f nucleases and introducing Cas12f-MG119-28 as a highly efficient ortholog with potential therapeutic applications. However, several critical aspects warrant further discussion and investigation:

1. Comparative scope and phylogenetic breadth: The title suggests a comparative characterization of Cas12f orthologs, yet the study primarily focuses on three variants (OsCas12f, RhCas12f, and Cas12f-MG119-28). Other well-characterized orthologs (e.g., AsCas12f, Un1Cas12f, CnCas12f) are mentioned but not systematically compared. Including additional orthologs would strengthen the study's conclusions and provide a more comprehensive evolutionary perspective. A phylogenetic analysis (structure-based is preferred) incorporating more Cas12f variants could reveal whether the observed structural and mechanistic features (e.g., extended REC helices in Cas12f-MG119-28) are evolutionarily conserved or lineage-specific.

2. Evolutionary and functional implications: The study identifies extended helices in the REC domain of Cas12f-MG119-28, which enhance dimer stability and NTS-DNA/gRNA binding. Are there other natural Cas12f orthologs with similar structural adaptations? If so, do they also exhibit high editing efficiency? Testing additional orthologs with predicted REC domain extensions could validate whether this mechanism is a generalizable feature of highly active Cas12f nucleases.

3. Functional validation of structural findings: While mutational analyses support the role of the REC domain in dimer stability, the asymmetric dimer architecture complicates interpretation. A single mutation (e.g., R139A or F143A) may disrupt multiple interactions (gRNA, NTS DNA. Supplementary Figure 10), making it difficult to assign causality. Covalent dimer experiments (as demonstrated in Supplementary Figure 12) could be expanded to dissect the contributions of specific interactions (e.g., gRNA vs. NTS DNA binding) to editing efficiency.

4. Specificity and spacer length requirements: Recent studies (e.g., PMID: 40335752) show that REC domain insertions in IscB enhance DNA/gRNA stabilization and editing specificity. Given that Cas12f-MG119-28 has a large REC domain, does it influence spacer length requirements for efficient editing, or editing specificity compared to other Cas12f variants? A side-by-side comparison of editing specificity (e.g., using GUIDE-seq) would clarify whether the extended REC domain confers additional targeting precision.

5. Performance in mammalian cells for therapeutic potential: Miniature Cas12f nucleases are attractive for AAV delivery, but their efficacy relative to classical Cas9/Cas12a systems remains generally lower. A direct comparison of Cas12f-MG119-28 with SpCas9, AsCas12a, or LbCas12a in mammalian cells (e.g., indel efficiency and off-target rates) would better establish its therapeutic applicability.

Version 2:

Decision Letter:

Our ref: NSMB-A51252B

4th Feb 2026

Dear Dr. Taylor,

Thank you for submitting your revised manuscript "Comparative characterization of Cas12f orthologs reveals mechanistic features underlying enhanced genome editing efficiency" (NSMB-A51252B). It has now been seen by the original referees and their comments are below. The reviewers find that the paper has improved in revision, and though concerns on behalf of Reviewer #3 remain with respect to whether Al3Cas12f has been sufficiently benchmarked, also versus mainstream editors like spCas9, concerns which we find valid but addressable in follow-up studies and not diminishing the interest in this particular study, we can now accept the manuscript in principle for publication in Nature Structural & Molecular Biology, also given the explicit support of the other two reviewers, pending minor revisions to satisfy the referees' final requests, particularly with respect to the need for future further benchmarking, and to comply with our editorial and formatting guidelines.

We are now performing detailed checks on your paper and will send you a checklist detailing our editorial and formatting requirements in about 1-2 weeks. Please do not upload the final materials and make any revisions until you receive this additional information from us.

To facilitate our work at this stage, it is important that we have a copy of the main text as a word file. If you could please send along a word version of this file as soon as possible, we would greatly appreciate it; please make sure to copy the NSMB account (cc'ed above).

Sincerely,

Dimitris Typas
Senior Editor
Nature Structural & Molecular Biology
ORCID: 0000-0002-8737-1319

Reviewer #1 (Remarks to the Author):

The authors have addressed most of my concerns, and I support the publication of the manuscript. However, I suggest a few minor points to improve the clarity and readability before publication.

Points:

Fig. 1b: The font size for the sgRNA nucleotides should match that of the TS to clarify RNA-DNA base pairing. It would be better to specify "NNNNN" in magenta (likely the PAM positions).

Fig. 3a: Nucleotide sequences are difficult to discern. Using lighter background colors (magenta and blue) would improve visibility.

Fig. S1: The scheme for the in vitro DNA cleavage experiments is still difficult to understand. The authors cleaved the target library DNA with all possible 8N PAMs using the nucleases and then treated the products with Klenow Fragment or Mung Bean Nuclease. Subsequently, the products were ligated with NGS adapters and amplified by PCR. As described in lanes 489-492, active systems that successfully cleaved the PAM library yielded bands around 188 or 205 bp in an agarose gel when blunted with Klenow Fragment and a 195 bp band when blunted with Mung Bean Nuclease. Therefore, uncleaved target plasmids are not visible due to their low amounts, while only PCR-amplified cleaved fragments are observed on the gel; is this correct? It would be helpful to add information about the size of the library plasmid in the Methods section and include a schematic illustrating the experimental procedure in Fig. S1.

Reviewer #2 (Remarks to the Author):

My previous concerns have been fully addressed.

Reviewer #3 (Remarks to the Author):

The authors have addressed some of my concerns. However, two critical issues remain unanswered: the specificity of cleavage and the relative activity of Al3Cas12f compared to the widely used SpCas9 and AsCas12a (or LbCas12a). Specificity is a key metric in gene editing, which may not be the central focus of this study, but preliminary characterization would significantly enhance the research. It is often observed that high activity tends to correlate with lower specificity; however, with structural insights, protein engineering may enable both high activity and high specificity. Second, since Al3Cas12f is proposed as a highly active gene-editing enzyme, it is essential to conduct a direct side-by-side comparison with SpCas9 and AsCas12a (or LbCas12a). This will help readers understand the relative activity of Al3Cas12f, particularly whether the RKK mutant achieves activity comparable to that of SpCas9 and AsCas12a (or LbCas12a). Such a comparison can preliminarily assess whether it truly meets the application requirements for basic research technologies or even for gene therapy applications.

Version 3:

Decision Letter:

4th Mar 2026

Dear Dr. Taylor,

We are now happy to accept your revised paper "Comparative characterization of Cas12f orthologs reveals mechanistic features underlying enhanced genome editing efficiency" for publication as an Article in Nature Structural & Molecular Biology.

Your paper will be published online soon after we receive proof corrections and will appear in print in the next available issue. You can find out your date of online publication by contacting the production team shortly after sending your proof corrections.

Authors may need to take specific actions to achieve compliance with funder and institutional open access mandates. If your research is supported by a funder that requires immediate open access (e.g. according to <https://www.springernature.com/gp/open-science/plan-s-compliance> Plan S principles or the <https://www.springernature.com/gp/open-science/us-federal-agency-compliance> NIH public access policy) then you should select the gold OA route, and we will direct you to the compliant route where possible. Because authors warrant under our subscription licensing terms that they haven't committed to licensing any version of their article under a licence inconsistent with the terms of our agreement – including the applicable embargo period – publication under the subscription model isn't suitable for authors whose funders require no embargo.

Sincerely,

Dimitris Typas
Senior Editor
Nature Structural & Molecular Biology
ORCID: 0000-0002-8737-1319

Response to Reviewer #1:

CRISPR-Cas12f nucleases are smaller than other CRISPR enzymes and have been used as miniature genome editors. In this study, Guan et al. identified Cas12f-MG119-28, which shows robust genome editing activity in human cells. Furthermore, the authors performed structural and functional analyses of Cas12f-MG119-28, *Oscillibacter* sp. Cas12f (OsCas12f) and *Ruminiclostridium herbifermentans* Cas12f (RhCas12f). A comparative structural analysis reveals both similarities and differences among these orthologs, providing insights into key mechanistic determinants of Cas12f activity. Overall, this manuscript advances our understanding of the diverse miniature Cas12f nucleases and offers a framework for developing genome-engineering tools with therapeutic potential. However, the quality of some figures is extremely poor and cannot be properly interpreted. Therefore, I am unable to support the publication of the current version of the manuscript. The authors should address the following points for substantial revision.

We thank the reviewer for recognizing the insights into key mechanistic determinants of Cas12f activity provided by our manuscript. Like the reviewer, we believe that our work could provide a framework for downstream engineering of better nucleases. We thank the reviewer for the constructive feedback on figure quality and clarity. We also thank the reviewer for delineating new experiments to support our claims. We believe that the experiments that were performed to address these comments have significantly strengthened the manuscript.

Major points:

L84: “These results, combined with the observation that two target sites achieved >90% 85 editing at AAVS1 (Fig. 1d), suggest that this nuclease is highly active in its native form.” - This is interesting and surprising. However, the authors should more carefully evaluate the editing activity of Cas12f-MG119-28 by comparing it with other highly active nucleases. At minimum, they should compare its editing efficiency with those of SpCas9 and enAsCas12 (10.1016/j.cell.2023.08.031) across multiple sites in human cells.

We agree with the reviewer that benchmarking Al3Cas12f (please note that Cas12f-MG119-28 has been renamed to Al3Cas12f) against other highly active nucleases is important for comparison. We performed genome editing assays using two enzymes available to us: LbCas12a and OsCas12f and compared their editing efficiencies to Al3Cas12f at eight different loci (see new **Fig. 1d, e**). Briefly, our results show that Al3Cas12f has higher activity than OsCas12f at 6 out of 8 loci and significantly outperforms LbCas12a at 2 out of 8 loci.

We have also updated the manuscript to include this comparison at L88-100:

“We next studied the activity of Al3Cas12f in human cells. Results from a gRNA screen targeting intron 1 of the ALB gene, exon 3 of the *APOA1* gene, and AAVS1 site within *PPP1R12C* intron 1 showed that 27 target sites displayed >10% editing, 19 sites displayed >50% editing, and 10 sites displayed > 90% editing across AAVS1 and *APOA1*

(assessed as the percentage of reads from next-generation amplicon sequencing that contained insertions or deletions) (**Fig. 1c**). In addition to screening Al3Cas12f across multiple loci we also tested different spacer lengths to optimize Al3Cas12f cleavage activity. Results revealed that a 19 nt spacer length seemed optimal across multiple sites, with the AAVS1-F1 gRNA showing similar editing efficiencies across 19-21nt spacers (**Suppl. Fig. 5**). Together, these results suggest that Al3Cas12f is highly active in its native form. To further validate the high levels of activity observed in mammalian cells, we compared the activity of Al3Cas12f relative to other CRISPR-Cas orthologs. Al3Cas12f outperformed OsCas12f across 6 out of 8 loci and exhibited significantly higher activity than LbCas12a at two sites (*NLRC4* and *NUDT16*) (**Fig. 1d**)."

L314: "While we show the gRNA of Cas12f-MG119-28 has already been naturally optimized..."

- It would be beneficial to further improve the editing activity of Cas12f-MG119-28 through structure-guided engineering.

We agree that structure-guided engineering is an important next step in the optimization of the Al3Cas12f as a gene editing tool. The original submission focuses on identifying the features that make the wild type Al3Cas12f an efficient genome editor in cells. However, we took the opportunity during the revision period to perform some structure-guided engineering. Please refer to the new Result section titled "**Structure-guided engineering of Al3Cas12f increases editing efficiency**" in the main text as well as the new **Fig. 7**, and **Suppl. Figs. 16 and 17**. Overall, the engineering efforts led to all triple mutant variants exhibiting levels of editing efficiency that significantly outperformed the wild type enzyme. Excitingly, one triple mutant exhibited saturating editing efficiency at a site where the wild type had ~40% efficiency. The following text was included at L311-334:

"Despite saturating levels of editing efficiency at AAVS1, WT Al3Cas12f showed < 61% editing at other loci (e.g., *SOD1*, **Suppl. Fig. 5**). To improve the activity of Al3Cas12f, enzyme variants were designed based on the cryo-EM structure in combination with phylogenetic sequence analysis to determine functionally conserved residues. This approach focused on increasing the positive charge of residues within hydrogen-bonding distance to the DNA substrate and/or the gRNA. Most of these residues were in the Wedge (WED) domain (**Suppl. Fig. 16**). Because Al3Cas12f exists as an asymmetric homodimer, special care was taken to ensure that the intended mutation would not disrupt (1) residues critical for the dimerization of the two protomers or (2) key protein-nucleic-acid interactions necessary for binding.

We started by testing single point mutations and evaluating the editing efficiency of these variants. Six single point mutation variants with editing efficiency over 60% were chosen to create combinatorial mutations (**Suppl. Fig. 17a**). We tested 35 double- and triple mutant variants in K562 cells targeting one AAVS1 site and the *SOD1* sites as well as one hexamutant that combined six mutations (**Suppl. Fig. 17b and Fig. 7a**). 21 of the 35 combinatorial mutants showed at least 75% editing efficiency at *SOD1* with the A9 gRNA with a quarter of the dose used in the previously described spacer length assay, 125 ng mRNA and 50 pmol of gRNA (**Fig. 7b**). Only the hexamutant displayed lower

levels of editing efficiency (~10%) than the wild type (~40%). From this assay the K79R/M190K/E222K variant (# 53) was chosen as the lead AI3Cas12f variant, hereby referred to as AI3Cas12f RKK (**Fig. 7a, b** and **Suppl. Data. 1**). Furthermore, about half of the protein variants led to at least 90% editing efficiency at the AAVS1 locus (**Suppl. Fig. 17b**). AI3Cas12f RKK was benchmarked against wild type enzyme using eight reference guides previously employed for AsCas12f 21. Six of these guides display >80% Indels, with up to a 26-fold increase in activity at the *EMC6* locus (**Fig. 7c**). Overall, these results show that AI3Cas12f RKK is a highly efficient and compact nuclease that could be further optimized for genome editing.”

Since the engineering of AI3Cas12f represent significant new data in the manuscript, we also included remarks in the Abstract at L32-35:

“Leveraging these structural insights, we generated an engineered AI3Cas12f variant (RKK) that increases editing and improves activity across multiple genomic loci. By overcoming locus-dependent variability and an apparent potency threshold, this engineered compact editor expands the feasibility of low-dose, AAV-compatible therapeutic genome editing.”

These results were also incorporated into the Discussion section at L386-393:

“WT AI3Cas12f showed substantial locus-to-locus variability, raising the likelihood of insufficient editing at a given disease-relevant target. To address this potency limitation, we performed structure-guided engineering and generate an engineered variant, AI3Cas12f RKK, that increased average editing from <10% to >80% (under the tested conditions). This broad improvement across loci helps overcome an activity threshold needed for development as a therapeutic gene-editing system. Consistent performance across multiple loci supports AI3Cas12f RKK as a more generally deployable platform, and its high activity at lower input doses may be advantageous for improving the safety-efficacy balance in clinical translation.”

Points:

L71: “Upon inspection of the genomic regions encoding the nuclease genes, we identified the corresponding tracrRNAs and crRNAs. Cas12f-MG119 representatives require large sgRNAs (129 - 156 nt) as empirically confirmed (Fig. 1b, Suppl. Fig. 1 and Suppl. Data 2).”

- Fig. 1b and Suppl. Fig. 1 lack sufficient information and should be substantially revised. In addition, the sequences of the crRNA and tracrRNA should be provided.

Fig. 1b was removed since it was redundant with **Fig. 3a**. The legend was also updated.

Suppl. Fig. 1 was revised following suggestions by the reviewer (below), and the legend now includes experimental details.

The sequences of the sgRNAs were separated into crRNA (including the GAAA tetraloop) and tracrRNA (**Suppl. Data. 1**) and **Fig. 3**.

L74: “We determined that these systems recognize T-rich PAMs using an in vitro cleavage assay of an 8N PAM library (Fig. 1c), and cut the target strand at 20, 22, and 23 nt from the 5’ PAM sequence and the non-target strand at 11 and 13 nt (Suppl. Fig. 2 and 3).”

- Since CRISPR nucleases use guide (spacer) segments of various lengths (e.g., 20 nt for SpCas9), the authors should specify the guide length of Cas12f-MG119-28 and examine how guide length affects DNA cleavage efficiency in vitro.

The spacer length used in the assay was 20 nt, and it is now mentioned in the **Fig. 1** legend.

Additionally, we are providing a new **Suppl. Fig. 5**, demonstrating the effect of spacer length on editing efficiency at two different loci (*AAVS1* and *SOD1*) in K562 cells. Depending on the site targeted at each locus, the 19 nt spacer displayed the highest levels of editing efficiency, suggesting a preference of the system for this spacer length.

L102: “This enzyme most closely resembles the structure of *Clostridium novyi* (CnCas12f)”
- Including a figure comparing the structures of CnCas12f and Cas12f-MG119-28 would be helpful.

We agree with the reviewer that this provides a more thorough comparison between Al3Cas12f and CnCas12f. **Suppl. Fig. 8** now includes a side-to-side comparison of the structures and their REC domains.

L109: “Since there are limited Cas12f structures available, we aimed to perform a more thorough comparison of several orthologs known to have high levels of genome editing activity, obtaining structures of OsCas12f and RhCas12f bound to their respective guides and target DNA”

- The rationale for selecting OsCas12f and RhCas12f for structural and functional comparison should be explained. Also, the sequence identities between Cas12f-MG119-28 and these orthologs should be provided.

We agree with the reviewer that the rationale should be clearer. The following statement has been added to the manuscript at L121-125:

“A recent publication highlighted the use of OsCas12f and RhCas12f as potential genome editing tools. Therefore, we reasoned that obtaining structures of OsCas12f and RhCas12f bound to their respective gRNAs and target DNA might provide more insights into mechanism of Cas12f nucleases.”

L124: “their RNA is much shorter than their counterpart since they contain only four stem loops”

- Stem 1 is not classified as a stem loop; clarify this point.

We thank the reviewer for pointing out the issue of Stem 1 nomenclature. Previous Cas12f publications including Su *et al.*, 2024, Hino *et al.*, 2023, and Takeda *et al.*, 2021, call it

Stem 1. Accordingly, we use the same nomenclature in all of our structures.

L194: “Cas12f-MG119-28 displayed two different intermediates, both with a fully formed 20 bp R-loop: a pre-product state prior to RuvC.2 docking (State I) and a post-product state with a fully docked RuvC.2 domain at the PAM-distal end of the spacer (State II)”
- Do these states correspond to pre- and post-cleavage states? Clarify this.

Since RuvC.2 docking is required for cleavage, we believe that State I is in the pre-cleavage state. For State II, the target strand (TS) and non-target strand (NTS) are not resolved at the cleavage site. Since the density in this region is not well resolved and these structures were taken after pre-incubating the sample for 30 minutes, our kinetic results would indicate that State II is a post-cleavage structure, but we do not have direct evidence in the structure.

L213: “We show that this lid is essential for cleavage by mutating E334-R343 to alanine, preventing the lid from undergoing the rearrangements necessary for cleavage”
- Why does the alanine mutation in E334-R343 prevent the rearrangement of the lid region?

Similar experiments were previously performed with the UnCas12f structure (Xiao *et al.*, 2021). One possible explanation is that the lid region facilitates proper positioning of the DNA substrate within the active site. In the alanine substitution mutant, DNA could not be positioned properly, resulting in severe defects in DNA cleavage activity.

L223: “In contrast to OsCas12f, both structures of Cas12f-MG119-28 showed the lid in an open conformation, indicating that Cas12f-MG119-28 can adapt its cleavage-active conformation prior to RuvC.2 docking.”
- It would be helpful to include an explanation for why, unlike OsCas12f, Cas12f-MG119-28 can adopt an active conformation before RuvC.2 docking.

Al3Cas12f can form a full 20 bp R-loop prior to RuvC.2 docking. One possible explanation for this is the increased contacts in the REC domain with the NTS, which should make the formation of the R-loop more energetically favorable. In all our Cas12f structures the RuvC lid sterically occludes the pathway that the DNA:RNA heteroduplex takes to form the full R-loop. Since the formation of the R-loop then allows for the RuvC lid to rearrange, we believe that the stabilization of the NTS by the REC domain (see F102A mutations in **Suppl. Fig. 12b**) allows for full R-loop formation without requiring the RuvC.2 rearrangement, allowing Al3Cas12f to adapt the cleavage-active conformation.

L265: “For RhCas12f, approximately 45% partitioned towards non-target strand cleavage, essentially showing little preference towards which strand is cleaved first.”
- How can RhCas12f cleave the target strand prior to the non-target strand? Please provide structural explanations.

Since both RuvC domains are flexible in this structure, we speculate that RuvC.1 can rearrange more freely to cleave both strands without significant specificity towards one or the other.

A short sentence has been added at L288-290 for structural explanation:

“Structurally, RuvC.1 domain remain highly flexible in RhCas12f, suggesting it may rearrange more freely to cleave both strands without significant specificity towards one or the other.”

Fig. 1b: This figure is of very poor quality and lacks clarity. Please show the sequences and indicate the spacer and repeat regions of the crRNA and the tracrRNA region.

We apologize for this oversight. As mentioned in response to the comment above, this figure has been removed. The sequence and regions for tracrRNA, crRNA, and tetraloop can be found in **Fig. 3a**.

Fig. 1c: Unnecessary labels (such as 119-28-U40... etc.) should be removed for clarity.

Unnecessary labels have been removed.

Also note that the Seqlogo with the U67 spacer was removed and replaced by a schematic of the cut sites. This was done for consistency with the data presented for Cas12f-MG119-1, -2 and -3 in **Suppl. Fig. 3**, upon request by the reviewer. Likewise, the AI3Cas12f NTS Seqlogo was moved to **Suppl. Fig. 3**.

Fig. 1d: Did the authors measure editing efficiency only once? Replicates (triplicates or more) are needed for statistical validity.

We have now included results (and updated methods) for an assay conducted to replicate the measures of editing efficiency previously shown in **Fig. 1c**, including guides in triplicate. The legend was also updated to capture these changes. We have updated the spacer sequences in Suppl. Data. 1 to reflect the new data in triplicate. Please note that the list of guides tested in triplicate differs from the original figure: (# 13, 19, 21, 28, 30, and 33) were removed and others were added (# 34 - 37). The conclusion of the figure is unchanged, but now the data span only two loci instead of three.

Lines 530-544 in the Methods section now read:

“Spacer sequences targeting human loci were identified by searching the Albumin intron 1, *APOA1* exon 3, and *AAVS1* loci for "TTR" PAMs to reduce the number of gRNAs tested for an initial screen. gRNAs were designed by appending the adjacent 20 nt regions to the 152 nt gRNA scaffold. After an initial gRNA screen with a total of 104 gRNAs, 30 of them were selected for a second round of validation (**Suppl. Data. 1**), appending the spacer sequences to the minimal 134 nt gRNA scaffold. The top 27 gRNAs (n= 3) targeting *AAVS1* and *APOA1* are shown in **Fig. 1c**. For nucleofection with AI3Cas12f mRNA, 500 ng mRNA and 200 pmol of gRNA were mixed together and incubated on ice until cells were prepared.

Two spacers that had previously shown < 50% editing efficiency were chosen to assess the performance of the AI3Cas12f engineered variants. For nucleofection with wild

type and combinatorial mutant AI3Cas12f mRNA and *SOD1* A9 gRNA or AAVS1 F4 gRNAs (134 nt gRNA scaffold and 19 nt spacer, **Suppl. Data. 1**), 125 ng mRNA and 50 pmol of gRNA were mixed together and incubated on ice until cells were prepared, which is a quarter of the dose initially tested with single point mutants at both loci. This dose was chosen to enable the assessment of a fold change in editing efficiency regardless of saturating levels in some cases.”

Fig. 1e: Compare the activity of MG119-28 with that of enAsCas12f (10.1016/j.cell.2023.08.031) in *E. coli*. Also, ensure consistent notation: “As. Cas12f” etc. should be correctly written as “AsCas12f.” etc.

The original GFP depletion assay performed in *E. coli* only used a single target, while multiple targets may be necessary to compare their activity more properly (as raised by Reviewer #2). Nevertheless, we decided to compare the editing efficiency of AI3Cas12f with LbCas12a and OsCas12f in human cells (see **Fig. 1d, e**). We believe that these results are more appropriate for comparison. Therefore, we have only included these data in the final manuscript. Please see response to the first major point where we compared the editing efficiency of AI3Cas12f to two other CRISPR-Cas nucleases.

Notation has been corrected.

Fig. 2a: It would be better to include domain schematics of Cas12f-MG119-28, OsCas12f, and RhCas12f, with their residue numbering, highlighting similarities and differences.

We agree with the reviewer, we have modified **Fig. 2** and added domain organization for each of the orthologs presented in this manuscript.

Fig. 2b-d: Show the protein names for clarity.

We agree with the reviewer, we have added the protein name next to the domain organization.

Fig. 3a: Indicate the crRNA and tracrRNA regions, as well as the tetraloop linking them.

We now indicate the crRNA, tracrRNA, and tetraloop using different colors and have labeled the regions.

Fig. 3b: Add labels for the protein domains.

Since the purpose of this figure is to highlight the presence and shape of the sgRNA, we believe that adding labels to the domains detracts from the purpose of the figure.

Fig. 3c: Depict hydrogen bonds using dashed lines for clarity.

Dashed lines have been added for clarity.

Fig. 4f: Clarify how each panel relates to those in e. Currently, it is confusing that b, d, and f contain multiple panels. Depict hydrogen bonds with dashed lines.

We agree with the reviewer. We have added individual panels in a revised **Fig. 4** to increase the clarity of different regions of the REC domain.

Supplementary Figure 1: The figure quality is very poor and not interpretable. Please indicate what each band represents (DNA substrate, cleavage product, sgRNA). Include no-enzyme control. Add schematics of sgRNA designs. Briefly explain the reaction conditions (incubation time, PAM sequences, guide lengths, visualization methods). Clarify what labels like “U50_A2” mean and explain the unlabeled lanes. Are the procedures described in the method section?

We apologize for the oversight and appreciate the suggestions. The bands corresponding to DNA cleavage have been labeled.

A no guide negative control (Al3Cas12f nuclease only) was included in this experiment.

No sgRNAs schematics were included, as we believe it would be overly distracting for readers. However, the sgRNA sequences are available to the readers in **Suppl. Data. 1**.

We confirm that reaction conditions were included in the methods section, under “In vitro cleavage reactions to enable PAM determination”. Key elements from the methods section have now been included in the figure legend.

Unnecessary parts of the labels (well #: ‘_A2’) were removed.

We are also submitting the raw gel images in **Source Data File**.

Supplementary Figure 2: This figure is also of poor quality; please clarify what each band represents. Remove unnecessary lanes and present upper and lower parts separately. Briefly explain reaction conditions in the legend.

We apologize for the oversight and appreciate the suggestions.

Bands were labeled to clarify that they represent cleavage products in the target and non-target strands. In addition, the image was cropped to remove unnecessary lanes, and the upper and lower parts were separated from each other. We are also submitting the raw gel images in **Source Data File**.

As with **Suppl. Fig. 1**, we confirm that reaction conditions were included in the methods section, under “In vitro cleavage reactions to enable PAM determination”. Key elements from the methods section have now been included in the figure legend.

Supplementary Figure 3: Explain what the two PAM logos in panels a and b indicate. Why is only one panel shown in c? Include schematics of guide (spacer) sequences, target DNA, and PAMs, along with cleavage sites. Clarify the guide lengths used in the assay.

The PAM sequence was obtained by NGS sequencing of cleavage products treated with the DNA polymerase I large Klenow fragment or Mung Bean nuclease, to confirm the PAM sequence from the target strand and the non-target strand, respectively. The corresponding SeqLogo is shown for each cleavage product by nuclease in panels a, b, c, and d (details added to the figure legend).

In the case of Cas12f-MG119-3 (panel c), less than a 1000 reads (below threshold) were obtained from the cleavage product treated with Mung Bean nuclease. Therefore, no SeqLogo or cut site histograms were generated for the NTS, and only the TS data is shown.

For consistency, the AI3Cas12f SeqLogo for the NTS, as histograms of the number of reads mapped to the cleavage site on each strand, are now included in this figure (panel d). The SeqLogo for the TS is shown in **Suppl. Fig. 1c**.

Schematics of guide bound to target DNA (the U40 spacer sequence) were included, along with the cleavage sites indicated with black arrows. Additionally, a diagram of the spacer and target DNA showing the cut sites for AI3Cas12f was included in **Fig. 1b**

The length of the sgRNAs (without spacers) used in the assay are included in the legend of **Suppl. Fig. 2**; the sgRNA sequences can be found in **Suppl. Data. 1**, and the U40 spacer sequence is included in the Methods section.

Supplementary Figure 3 and 4: Merge these figures. The SEC elution profile should be presented before the SDS-PAGE gel. In the SEC data, please label the molecular masses of standards and indicate fractions analyzed by SDS-PAGE.

These figures were merged (as **Suppl. Fig. 4**). The SEC elution profile is presented before the SDS-PAGE gel.

The molecular masses of the standards and the fractions analyzed by SDS-PAGE were labeled on the SEC elution profile.

Supplementary Figure 7: Enlarge labels for better readability

We have modified the size of the labels to allow for easier readability.

Minor points:

L50: “Although these proteins have yet to reach saturating levels of editing in mammalian cells, rational engineering of Cas12f proteins, as well as their guide RNA (gRNA) scaffolds, have significantly enhanced their gene-editing activity 21,23–25.”

- Please cite a recent publication on SpCas12f (<https://doi.org/10.1093/nar/gkaf588>).

Citation has been added.

L57: “we solved the structures of OsCas12f and RhCas12f, conducted kinetic analyses of all three orthologs...”

- Briefly describe OsCas12f and RhCas12f here, as the introduction is independent from the abstract.

We have added this phrase to increase clarity and introduce OsCas12f and RhCas12f at L63-65:

“We also studied two newly identified Cas12f orthologs that originate from *Oscillibacter sp.* (OsCas12f) and *Ruminiclostridium herbifermentans* (RhCas12f). Both orthologs exhibit high editing efficiency in human cells.”

L268: “In previous studies, we have performed magnesium-initiated cleavage experiments where the enzyme, guide RNA, and DNA are preincubated in the absence of magnesium and then magnesium is added to initiate the reaction.”

- Please include relevant references.

We have added Hibshman *et al.*, 2024 and Fregoso Ocampo *et al.*, 2025 as references.

L297: “Prior to this study, three structures of Cas12f nucleases had been described.”

- Include corresponding references.

We have modified this phrase and added references.

L299: “Structures of Cas12f-MG119-28, OsCas12f, and RhCas12f in combination with AsCas12f, UnCas12f, and CnCas12f...”

Citation has been added.

L478: “Approximately 1E+5 cells were transfected”

- Revise to standard notation, e.g., “1 × 10⁵ cells.”

This has been corrected.

L559: “For Cas12f-MG119-28, 11,080 movies were collected, and 2,039,236 particles were extracted for downstream analysis.”

- There are discrepancies with Supplementary Fig. 6, which indicates 2,039,247 particles. Please verify and correct for consistency.

The particle number has been corrected to 2,039,247 for consistency.

Materials and methods: There are many typos that should be corrected. For example, 100mM -> 100 mM; E.coli -> E. coli (italicized); MgCl₂ (2 should be in subscript); 18000 rpm should be specified as g-force (e.g., "g" units); "micro" (not u), "prime" and "minus sign" should be used correctly and consistently.

E. coli notations have been italicized and MgCl₂ has been modified with "2" added as a subscript. Rpm has been converted into "g" and other signs have been corrected.

Supplementary Figure 11: Please add labels indicating the domains, RNA regions, and DNA regions.

Modifications have been added.

Supplementary Figure 13: Expand the figure legend to include more detailed explanations.

Legend has been expanded to:

"Contours show the change in χ^2 as a function of values for individual rate The dashed line indicates the χ^2 threshold corresponding to the 95% confidence interval used for reporting rates."

Reviewer #2 (Remarks to the Author):

The manuscript presents a structural and functional characterization of three Cas12f orthologs, with a particular focus on the newly identified Cas12f-MG119-28 variant. The study combines cryo-EM structural determination with extensive biochemical and kinetic analyses to understand the mechanistic basis for the enhanced editing efficiency of Cas12f-MG119-28. The work provides valuable insights into CRISPR-Cas12f working mechanisms that could inform future engineering of compact genome editors. The manuscript is generally well-written and well-organized, though I have several suggestions for improvement as detailed below.

We would like to thank the reviewer for recognizing the value and impact of our manuscript. We are also grateful to the reviewer for multiple suggestions to improve the manuscript. We have addressed a majority of these suggestions, and we believe the manuscript has been significantly strengthened.

Major Comments

1. The name "Cas12f-MG119-28" is unnecessarily complex and could hinder broader adoption. Consider a simplified designation to align with conventions for ortholog naming.

Revision of the taxonomic classification of the DNA contig encoding Cas12f-MG119-28, revealed that this nuclease originated from *Alistipes* sp. 58_9_plus (strain Z76), a member

of the Phylum Bacteroidota. Additionally, through phylogenetic analysis of representative Cas12f sequences in the literature (**Suppl. Fig. 19**), we determined that this sequence is closely related to Al1Cas12f1 and Al2Cas12f1 (84 - 97% identity). Considering this information, we propose the name Al3Cas12f.

The following was added to the Methods section at L432-436:

“In the case of Cas12f-MG119-28, the consensus taxonomy of all the proteins encoded in the contig was used to assign a taxonomic affiliation. Taxonomy of the predicted proteins was assigned with Kaiju v15⁴¹. Most proteins encoded in the contig in which Cas12f-MG119-28 was identified were classified as belonging to *Alistipes* sp. 58_9_plus (strain Z76) and following convention in the literature it was named Al3Cas12f.”

2. The editing efficiency of the newly identified Cas12f should be benchmarked against larger nucleases, as the authors claim that it is a highly efficient genome editor.

We thank the reviewer for the suggestion. We compared gene editing efficiency of Al3Cas12f to OsCas12f and LbCas12a in human cells. These experiments demonstrated that Al3Cas12f has higher activity than OsCas12f at 6 out of 8 loci and significantly outperforms LbCas12a at 2 out of 8 loci (see **Fig.1d, e** and results section).

3. Lines 80-84: The manuscript states that editing efficiencies were determined for multiple loci, but Figure 1d lacks error bars or indication of biological replicates. Given this is a key result supporting Cas12f-MG119-28's high activity, please clarify whether the data represents biological replicates.

The in-cell assay previously shown in **Fig. 1d** was repeated with guides in triplicate and it is now shown as **Fig. 1c**. Not only were the results replicated, albeit with the 134 nt sgRNA scaffold that was used to obtain the structure but also improved upon. Please see our response to reviewer # 1 (above) and the Methods section (“Nuclease mRNA production” and “Editing efficiency in mammalian cells”) for experimental details.

4. Lines 88-90: The comparison of editing efficiency in *E. coli* appears to use a single GFP reporter target. As target sequence preferences significantly influence Cas12f variant's activity, I recommend extending the comparison to include multiple target sequences to ensure observed performance differences are generalizable rather than target-specific.

We thank the reviewer for the suggestion, and we agree that multiple targets are necessary for these comparisons. To further benchmark Al3Cas12f against other highly active nucleases, we have included the data in **Fig. 1d**, which shows the direct comparisons between Al3Cas12f, OsCas12f and LbCas12a at eight different loci in human cells. We believe that this would be a more appropriate comparison. Therefore, the original GFP depletion assay performed in *E. coli* has been removed from the manuscript.

5. Lines 176-179: While alanine substitutions in the REC domain reduce editing activity, Figure 4g data alone cannot establish whether this results from impaired dimerization or

other functional disruptions. To strengthen causality, additional characterization (e.g., SEC of mutants to quantify dimer stability) would be valuable, as done for WT protein (Suppl. Fig. 5).

We agree with the review that dimer stability is very important to test. We purified AI3Cas12f as well as three variants that showed significantly decreased activity in the GFP reporter assay. We then performed mass photometry analysis of these proteins to test the ratio of dimer: monomer after incubation at 37 °C for 1h. The results show a higher dimer: monomer ratio in the WT protein compared to the mutants, indicating that dimerization is more stable in the WT protein. This suggests that; indeed, the mutants significantly decrease dimer stability. We have added these data as a new **Suppl. Fig. 12** and included the following in the main text at L189-192:

“To further evaluate the impact of these mutations, we examined how specific amino acid substitutions influence the stability of the apo Cas12f dimer using mass photometry. Our results revealed that disrupting the dimer interface markedly compromises dimer stability (**Suppl. Fig. 12a**).”

6. The manuscript presents structural and kinetic evidence for divergent activation mechanisms among Cas12f orthologs. Integrating these findings into a mechanistic model figure would improve conceptual clarity.

We agree with the reviewer. We have now added a cartoon mechanism figure. (see **Fig. 8** and Discussion section).

Minor comments

7. The protein sequences used in the fig. 1a need to be indicated.

The protein sequences were now included in **Suppl. Data. 1**.

8. Lines 197-198: The reason for assigning N-terminal to OsCas12f.1 and C-terminal to OsCas12f.2 in the covalent dimer should be clarified, as this impacts interpretation of activation mechanisms.

We annotated all Cas12f structures and models based on previous annotations from published literature. Consistent with previous literature that has cloned and characterized these covalent dimers, molecule 2 contains a deletion in the RuvC linker that prevents association with the sgRNA. This allows us to distinguish between both monomers. Functionally, the monomer with the deletion will always correspond to molecule 2. Following the standard set in previous literature, we designed the ORF to contain the deletion in the C-terminal monomer, which is the way it is represented in the figures.

9. The unresolved RuvC domains in RhCas12f.1 resemble observations in TnpB (PDB: 8EX9) and Cas12n (PDB: 9J09). A comparative analysis of these systems'

conformational flexibility in the discussion section could provide insights into evolutionary constraints governing nuclease domain stability in compact Cas12 systems.

We thank the reviewer for this suggestion, and we believe that this could certainly be very interesting for a commentary or review article. However, as our manuscript studies different Cas12f orthologs, we have chosen to keep the focus on these enzymes.

10. Line 613: “intubated” should be corrected to “incubated”.

Correction has been made.

Reviewer #3 (Remarks to the Author):

The study investigates the structural and functional diversity of Cas12f nucleases, focusing on a newly identified ortholog, Cas12f-MG119-28, discovered through metagenomic analysis. The authors compare this nuclease with two other Cas12f variants (OsCas12f and RhCas12f) using cryo-EM, biochemical assays, and kinetic analyses. Four key findings are presented:

1. High editing efficiency: Cas12f-MG119-28 exhibits robust genome editing in human cells, and outperforms other Cas12f orthologs in E.coli.
2. Structural insights: Cryo-EM structures reveal distinct dimerization interfaces, gRNA architectures, and RuvC domain dynamics among the orthologs.
3. Mechanistic diversity: Cas12f-MG119-28 achieves efficient R-loop formation via a stable dimer interface and optimized gRNA, while OsCas12f and RhCas12f rely on different activation mechanisms.
4. Kinetic analysis: Cas12f-MG119-28 shows faster, and more efficient DNA cleavage compared to OsCas12f and RhCas12f, attributed to its stable dimerization and pre-formed R-loop.

Overall, this study represents a significant advancement in the CRISPR field, providing deep mechanistic insights into Cas12f nucleases and introducing Cas12f-MG119-28 as a highly efficient ortholog with potential therapeutic applications. However, several critical aspects warrant further discussion and investigation:

We would like to thank the reviewer for their comments on the manuscript. We agree with the reviewer that the manuscript presents four main points and that this represents a significant advance in the field. However, we do agree that all sections of the manuscript could be improved by addressing the comments presented by the reviewer. We believe that these changes have significantly strengthened the manuscript.

1. Comparative scope and phylogenetic breadth: The title suggests a comparative characterization of Cas12f orthologs, yet the study primarily focuses on three variants (OsCas12f, RhCas12f, and Cas12f-MG119-28). Other well-characterized orthologs (e.g., AsCas12f, Un1Cas12f, CnCas12f) are mentioned but not systematically compared. Including additional orthologs would strengthen the study's conclusions and provide a more comprehensive evolutionary perspective. A phylogenetic analysis (structure-based

is preferred) incorporating more Cas12f variants could reveal whether the observed structural and mechanistic features (e.g., extended REC helices in Cas12f-MG119-28) are evolutionarily conserved or lineage-specific.

A phylogenetic tree of representative Cas12f sequences (**Suppl. Data. 1**) was built and provided as **Suppl. Fig. 19**. The tree includes closely related sequences in the AI3Cas12f clade (please note that Cas12f-MG119-28 has been renamed to AI3Cas12f) (**Suppl. Data 1**), such as homologs Ti1Cas12f1, AI1Cas12f1 and AI2Cas12f1 (**Suppl. Data. 1**); as well as more distant homologs such as CnCas12f1, AsCas12f1 and SpCas12f1. Analysis of the multiple sequence alignment behind the tree revealed high diversity in a particular region of the Rec domain of these proteins (gray box, **Suppl. Fig. 20**).

Furthermore, the 3D structure of key representative systems previously characterized in the literature was obtained from the PDB. In cases where the 3D structure was not available (e.g., Cas12f-MG119-3 and AI1Cas12f1), it was predicted by folding the protein with the corresponding sgRNA using Boltz2 (a tool for modeling bimolecular interactions). The structures were aligned to the AI3Cas12f cryo-EM structure and the Rec domains analyzed (**Suppl. Fig. 19**). The results of this analysis indicated that alpha helices 2 and 3 in the Rec domain of AI3Cas12f may be a unique feature of the AI3Cas12f clade (for additional details, please refer to the **Supplementary Methods, Results and Discussion** in Supplementary Information).

2. Evolutionary and functional implications: The study identifies extended helices in the REC domain of Cas12f-MG119-28, which enhance dimer stability and NTS-DNA/gRNA binding. Are there other natural Cas12f orthologs with similar structural adaptations? If so, do they also exhibit high editing efficiency? Testing additional orthologs with predicted REC domain extensions could validate whether this mechanism is a generalizable feature of highly active Cas12f nucleases.

The analysis described above shows that alpha helices 2 and 3 that seem to be a distinguishing feature of the AI3Cas12f Rec domain are also present in other members of the clade (e.g., Cas12f-MG119-3, AI1Cas12f1, and AI2Cas12f1). However, when comparing editing efficiencies reported for other Cas12f systems in the published literature, AI3Cas12f seems to be highly active in K562 cells (saturating levels of editing efficiency, **Fig. 1c**), while AI1Cas12f1 appears to display lower efficiency in HEK293T cells (< ~ 35% editing efficiency; Sharrar et al. 2023) at different loci. Interestingly, the Rec domain of CnCas12f (and Cas12f-MG119-2) has longer alpha helices than AI3Cas12f (**Suppl. Fig. 19**). Nevertheless, WT CnCas12f does not display high levels of cleavage activity based on editing efficiency data reported in the literature (Su et al. 2023). Taken together, these findings suggest that the extended alpha helices in the Rec domain of AI3Cas12 may partly contribute to the high levels of editing efficiency displayed by this nuclease.

3. Functional validation of structural findings: While mutational analyses support the role of the REC domain in dimer stability, the asymmetric dimer architecture complicates interpretation. A single mutation (e.g., R139A or F143A) may disrupt multiple interactions (gRNA, NTS DNA. Supplementary Figure 10), making it difficult to assign causality.

Covalent dimer experiments (as demonstrated in Supplementary Figure 12) could be expanded to dissect the contributions of specific interactions (e.g., gRNA vs. NTS DNA binding) to editing efficiency.

Please note that AI3Cas12f residues has been renumbered by an offset of -41. Previous numbering erroneously included His and NLS tags at the N-terminal side. These tags are not included in the final sequences in Suppl. Data 1 for consistency with all other Cas12f variants. Therefore, AI3Cas12f was renumbered for consistency across all figures and supplementary data. PDB files have also been updated.

We agree with the reviewer that covalent dimer experiments are essential for determining the effect of different interactions. We have now cloned ten different covalent dimer mutants (5 different mutations one in each molecule) and tested these variants in the GFP reporter assay previously used to test these mutations. The results were added to **Suppl. Fig. 12**, and the following was added to the manuscript at Line 192-201.

“We then employed covalently linked dimers to assess how individual amino acid changes affect the GFP depletion activity of AI3Cas12f. Notably, the F112A, R63A, Y60A, and R49A substitutions, which disrupt key contacts within the dimer interface, led to a pronounced decrease in GFP depletion activity compared to wild type (**Suppl. Fig. 12b**). Meanwhile, the F102A substitution in monomer 1, which disrupts an important interaction with the NTS, exhibited significantly reduced ability in GFP depletion compared with the wild type. In contrast, the same mutation in monomer 2, which abolishes an interaction with the gRNA, caused only a modest reduction in activity (**Suppl. Fig. 12b**). Taken together, these findings demonstrate that a more stable dimerization interface significantly contributes to the enhanced nuclease activity by AI3Cas12f.”

4. Specificity and spacer length requirements: Recent studies (e.g., PMID: 40335752) show that REC domain insertions in IscB enhance DNA/gRNA stabilization and editing specificity. Given that Cas12f-MG119-28 has a large REC domain, does it influence spacer length requirements for efficient editing, or editing specificity compared to other Cas12f variants? A side-by-side comparison of editing specificity (e.g., using GUIDE-seq) would clarify whether the extended REC domain confers additional targeting precision.

It is unclear whether the large Rec domain of AI3Cas12 influences the spacer length requirements for editing efficiency. However, there is a preference for spacers of 19-nt in length as shown in **Suppl. Fig. 5**.

We did not compare the specificity of AI3Cas12f with other Cas12f variants, as we believe that testing the specificity of the enzyme could be the subject of an entire manuscript in its own right. Our study aims to test the structural basis for enhanced on-target activity, but we acknowledge that further testing is required to develop a fully characterized genome editing tool.

5. Performance in mammalian cells for therapeutic potential: Miniature Cas12f nucleases

are attractive for AAV delivery, but their efficacy relative to classical Cas9/Cas12a systems remains generally lower. A direct comparison of Cas12f-MG119-28 with SpCas9, AsCas12a, or LbCas12a in mammalian cells (e.g., indel efficiency and off-target rates) would better establish its therapeutic applicability.

We agree with the reviewer that benchmarking AI3Cas12f against other highly active nucleases is important. This has been included. Please see response to Reviewer #1 and #2. Briefly, we compared the editing efficiency of AI3Cas12f, OsCas12f and LbCas12a across eight difference loci. Our results showed AI3Cas12f has higher activity than OsCas12f at 7 out of 8 loci and significantly outperforms LbCas12a at 2 out of 8 loci (see **Fig. 1d, e**).

Reviewer #1 (Remarks to the Author):

The authors have addressed most of my concerns, and I support the publication of the manuscript. However, I suggest a few minor points to improve the clarity and readability before publication.

We thank reviewer 1 for recognizing that the modifications included in the revised version improved this manuscript.

Points:

Fig. 1b: The font size for the sgRNA nucleotides should match that of the TS to clarify RNA-DNA base pairing. It would be better to specify "NNNNN" in magenta (likely the PAM positions).

We have modified figure 1b by coloring the PAM magenta and changing it to the PAM that was used for all three enzymes in this study. The PAM is now denoted as "NNNNTTTG" with the TTTG highlighted in magenta. The font was also adjusted to show that indeed these regions are base pairing to each other.

Fig. 3a: Nucleotide sequences are difficult to discern. Using lighter background colors (magenta and blue) would improve visibility.

We agree with the reviewers that the nucleotides were hard to discern with the old coloring scheme. We have changed the colors of all nucleotide letters to white, which makes the figure significantly easier to interpret.

Fig. S1: The scheme for the in vitro DNA cleavage experiments is still difficult to understand. The authors cleaved the target library DNA with all possible 8N PAMs using the nucleases and then treated the products with Klenow Fragment or Mung Bean Nuclease. Subsequently, the products were ligated with NGS adapters and amplified by PCR. As described in lanes 489-492, active systems that successfully cleaved the PAM library yielded bands around 188 or 205 bp in an agarose gel when blunted with Klenow Fragment and a 195 bp band when blunted with Mung Bean Nuclease. Therefore, uncleaved target plasmids are not visible due to their low amounts, while only PCR-amplified cleaved fragments are observed on the gel; is this correct? It would be helpful to add information about the size of the library plasmid in the Methods section and include a schematic illustrating the experimental procedure in Fig. S1.

The reviewer is correct in their understanding. We understand the assay has many steps, so consistent with the reviewer's recommendation, we have included a schematic in Supplementary Figure 1 to better illustrate this protocol.

Reviewer #2 (Remarks to the Author):

My previous concerns have been fully addressed.

We thank reviewer 2 for their prior constructive comments on this manuscript.

Reviewer #3 (Remarks to the Author):

The authors have addressed some of my concerns. However, two critical issues remain unanswered: the specificity of cleavage and the relative activity of Al3Cas12f compared to the widely used SpCas9 and AsCas12a (or LbCas12a). Specificity is a key metric in gene editing, which may not be the central focus of this study, but preliminary characterization would significantly enhance the research. It is often observed that high activity tends to correlate with lower specificity; however, with structural insights, protein engineering may enable both high activity and high specificity. Second, since Al3Cas12f is proposed as a highly active gene-editing enzyme, it is essential to conduct a direct side-by-side comparison with SpCas9 and AsCas12a (or LbCas12a). This will help readers understand the relative activity of Al3Cas12f, particularly whether the RKK mutant achieves activity comparable to that of SpCas9 and AsCas12a (or LbCas12a). Such a comparison can preliminarily assess whether it truly meets the application requirements for basic research technologies or even for gene therapy applications.

We thank reviewer 3 for mentioning that we have addressed some of the concerns previously listed. To benchmark WT Al3Cas12f, we screened the activity of this system across eight different loci and compared it to the activity of LbCas12a and OsCas12f, which we had readily available. We agree with the reviewer that further studies should characterize the activity of the RKK mutant relative to other highly characterized orthologs.